# Interactions between deforestation, landscape rejuvenation, and shallow landslides in the North Tanganyika - Kivu Rift region, Africa

Arthur Depicker[1], Gerard Govers[1], Liesbet Jacobs[1], Benjamin Campforts[2], Judith Uwihirwe[3,4], and Olivier Dewitte[5]

[1]KU Leuven, Department of Earth and Environmental Sciences, Division of Geography and Tourism (Celestijnenlaan 200E, 3001 Heverlee, Belgium)

[2]CSDMS, Institute for Arctic and Alpine Research, University of Colorado at Boulder (Boulder, CO, USA)

[3]University of Rwanda, Department of Soil and Water Management, Faculty of Agricultural Engineering (Street KK 737, Kigali, Rwanda)

[4]Delft University of Technology, Faculty of Civil Engineering and Geosciences, Department of Water Management (Stevinweg 1, 2628 Delft, The Netherlands)

[5]Royal Museum for Central Africa, Department of Earth Sciences (Leuvensesteenweg 13, 3080 Tervuren, Belgium)

**Correspondence:** Arthur Depicker (arthur.depicker@kuleuven.be)

**Abstract.** Deforestation is associated with a decrease in slope stability through the alteration of hydrological and geotechnical conditions. As such, deforestation increases landslide activity over short, decadal timescales. However, over longer timescales (0.1-10 Ma) the location and timing of landsliding is controlled by the interaction between uplift and fluvial incision. Yet, the interaction between (human-induced) deforestation and landscape evolution has hitherto not been explicitly considered. We address this issue in the North Tanganyika-Kivu Rift region (East African Rift). In recent decades, the regional population has grown exponentially and the associated expansion of cultivated and urban land has resulted in widespread deforestation. In the past 11 Ma, active continental rifting and tectonic processes have forged two parallel mountainous Rift shoulders that are continuously rejuvenated (i.e. actively incised) through knickpoint retreat, enforcing topographic steepening. In order to link deforestation and rejuvenation to landslide erosion, we compiled an inventory of nearly 8,000 recent shallow landslides in © Google Earth imagery from 2000-2019. To accurately calculate landslide erosion rates, we developed a new methodology to remediate inventory biases linked to the spatial and temporal inconsistency of this satellite imagery. Moreover, to account for the impact of rock strength on both landslide occurrence and knickpoint retreat, we limit our analysis to rock types with threshold angles of 24-28°. Rejuvenated landscapes were defined as the areas draining towards Lake Kivu or Lake Tanganyika, and downstream of retreating knickpoints. We find that shallow landslide erosion rates in these rejuvenated landscapes are roughly 40 % higher than in the surrounding relict landscapes. In contrast, we find that slope exerts a stronger control on landslide erosion in relict landscapes. These two results are reconciled by the observation that landslide erosion generally increases with slope gradient and that the relief is on average steeper in rejuvenated landscapes. The weaker effect of slope steepness on landslide erosion rates in the rejuvenated landscapes could be the result of three factors: the absence of earthquake-induced landslide events in our landslide inventory, a thinner regolith mantle, and a drier climate. More frequent extreme rainfall events in the relict landscapes, and the presence of a thicker regolith, may explain a stronger landslide response to

deforestation compared to rejuvenated landscapes. Overall, deforestation initiates a landslide peak that lasts approximately 15 years and increases landslide erosion by a factor 2 to 8. Eventually, landslide erosion in deforested land falls back to a level similar to that observed under forest conditions, most likely due to the depletion of the most unstable regolith. Landslides are not only more abundant in rejuvenated landscapes but are also smaller in size, which may again be a consequence of a thinner

regolith mantle and/or seismic activity that fractures the bedrock and reduces the minimal critical area for slope failure. With this paper, we highlight the importance of considering the geomorphological context when studying the impact of recent land use changes on landslide activity.

## 1 Introduction

On steep terrain, the erosion caused by shallow landslides (with a maximal depth of a couple of meters) increases significantly as a result of deforestation (e.g. Montgomery et al., 2000; Mugagga et al., 2012). The removal of trees, due to either human or natural causes, decreases the slope stability through the alteration of hydrological and geotechnical conditions, such as the loss in soil cohesion due to tree root decay (Sidle et al., 2006; Sidle and Bogaard, 2016). After 10-20 years, depending on the climate and vegetation regeneration rate, this effect starts wearing off (Sidle and Bogaard, 2016). However, when forests are

permanently converted to grassland or cropland, the consequences of deforestation for landsliding can last much longer or even be permanent (Sidle et al., 2006).

While these general principles are well described, we do not yet fully understand the extent to which the response to deforestation is modulated by tectonic forcing, which typically occurs over timescales of 0.1-10 Ma (e.g. Whipple and Meade, 2006). A key distinction can be made here between actively incising, rejuvenating landscapes, in which landslides are a prime

slope-limiting mechanism and 'old', so-called relict landscapes, where hillslopes have had a long time to adapt to river incision (Burbank et al., 1996; Larsen and Montgomery, 2012). These two landscape types can be expected to respond differently to deforestation: in rejuvenating landscapes, hillslopes are already continuously adapting to river incision through landsliding (Egholm et al., 2013). In relict landscapes, on the other hand, hillslopes will slowly become less steep and landslides occur more sporadically, potentially allowing for a thick regolith mantle to develop (Schoenbohm et al., 2004; Bennett et al., 2016).

Climatic variations can also drive differences in landscape response to deforestation (Crozier, 2010), and in the context of litho-logically diverse landscapes, the effect of rock strength on both knickpoint retreat and landsliding must also be acknowledged (Parker et al., 2016; Baynes et al., 2018; Campforts et al., 2020).

Here, we aim to explore the interplay of deforestation and uplift-driven landscape rejuvenation on shallow landslide erosion. We focus our research on the North Tanganyika-Kivu Rift region (hereafter referred to as 'the NTK Rift', **Fig. 1**), part of

50 the western branch of the East African Rift. The area is characterized by frequent landsliding, mostly triggered by rainfall, widespread deforestation and active continental rifting (Hansen et al., 2013; Saria et al., 2014; Monsieurs et al., 2018a; Depicker

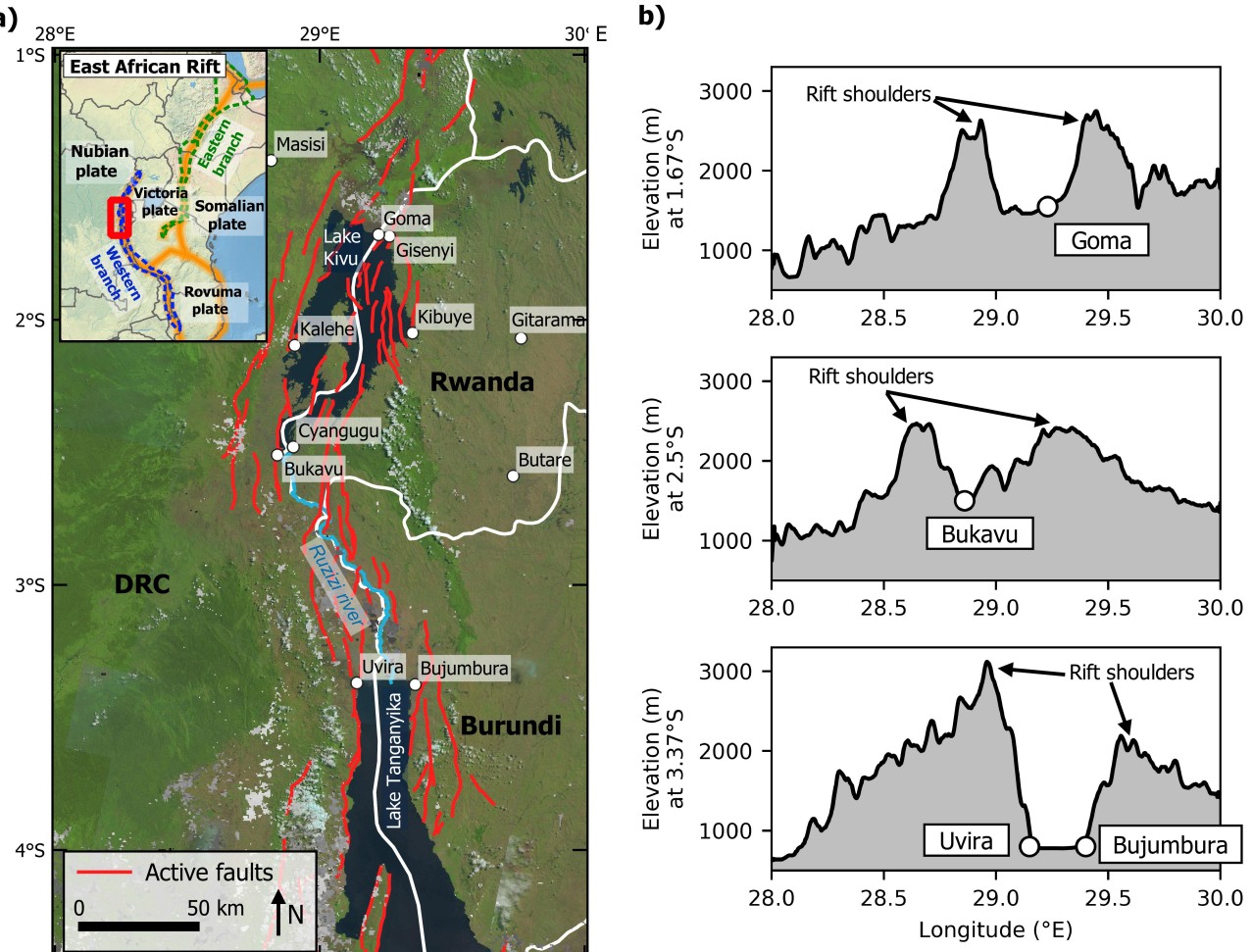

**Figure 1. Overview of the NTK Rift. a)** Extent of the studied area and active faults (Delvaux and Barth, 2010). LANDSAT-8 imagery is used as background (USGS, 2018). **b)** The transects of the elevation at different latitudes illustrate the elevated Rift shoulders, being the result of tectonic uplift. The four most populous cities of the NTK Rift (Goma, Bukavu, Uvira, Bujumbura) are located in between the Rift shoulders. The transect was derived from the 30 m resolution SRTM DEM (USGS, 2006).

et al., 2020; Dewitte et al., 2021). The study area is therefore an ideal setting to evaluate how deforestation affects landslide erosion in different landscape settings.

## 2    The North Tanganyika - Kivu Rift region

Active continental rifting in our study area is driven by the divergence of the Victoria and Nubia plates that started ca. 11 Ma ago and currently continues at a rate of ca. 2 mm/yr (Saria et al., 2014; Pouclet et al., 2016). Due to this setting, there is

widespread seismic activity, active volcanism, and uplift initiating landscape rejuvenation through knickpoint retreat (Smets et al., 2015; Delvaux et al., 2017; Dewitte et al., 2021). Adding to the geological complexity of the NTK Rift is the wide variability in age and strength of rock formations. The majority of rocks in the northern and eastern parts of the study area are of Mesoproterozoic age (1600-720 Ma), being mostly quartzites, granites, or pelites. The southwest is largely covered by either weathering-resistant quartzites or weathering-prone gneiss and micaschists of Paleoproterozoic age (2500-1600 Ma). Within the Rift shoulders, the same pattern of Meso- and Paleoproterozoic rocks is observed, save for the occurrence of much younger lithologies such as the river and lake sediments in the Ruzizi floodplain and the volcanic deposits (12 Ma - present) found around Bukavu and north of Goma (Delvaux et al., 2017; Laghmouch et al., 2018).

In the natural context, prior to widespread human activity, forests covered most of the DRC and the mountainous Rift shoulders in Rwanda and Burundi, while the vegetation transitioned to woodland savanna towards the east of our study area (Ellis et al., 2010; Aleman et al., 2018; Roche and Nzabandora, 2020). Only since the beginning of the 20[th] century, large scale deforestation has taken place, especially along the Rift shoulders and in Rwanda and Burundi (Ellis et al., 2010; Aleman et al., 2018). In 2000, the study area (ca. 88,500 km$^2$) had an estimated forest coverage of 73.1 %. Between 2001 and 2018, 4.5 % of these forests were cleared, mainly for the purpose of agriculture (Hansen et al., 2013; Tyukavina et al., 2018; Musumba Teso et al., 2019). Deforestation is therefore an indirect result of the fast-growing population, which increased from 89 inhabitants per km$^2$ in 1975 to 241 inhabitants per km$^2$ in 2015 (Hansen et al., 2013; JRC and CIESIN, 2015).

## 3 Methods

In the sections below, we first focus on the landscape characteristics of the NTK Rift that can exert a control on landslide erosion: forests, rejuvenation, rainfall, and rock strength. Next, we elaborate on the different aspects of the landslide erosion assessment: the compilation of an inventory and the calculation of shallow landslide erosion rates (in the context of the previously determined landscape characteristics).

### 3.1 Regional controls on landslide erosion

#### 3.1.1 Forest cover and deforestation

We characterize the NTK Rift in terms of forest dynamics by means of the global forest data presented by Hansen et al. (2013) (**Fig. 2a**, the data was updated in 2018). This dataset contains a tree cover map for the year 2000 and forest loss data for the period 2001-2017, both provided at a resolution of one arc-second (ca. 30 m). The tree cover data shows the percentage of tree coverage per pixel in 2000 and the forest loss data displays discrete values between 1 and 17, indicating the year in which deforestation took place. Based on these data, we distinguish three land cover classes: i) forest, having >25 % tree cover (as suggested by Hansen et al. (2013)), ii) deforested land, and iii) non-forest land, with ≤25 % tree cover. Both deforested and non-forest land encompass land use classes such as bare land, cropland, grassland, and urban land. Historically, current 'non-forest´ land used to be either savanna grassland or forest (Roche and Nzabandora, 2020). The difference between 'non-forest'

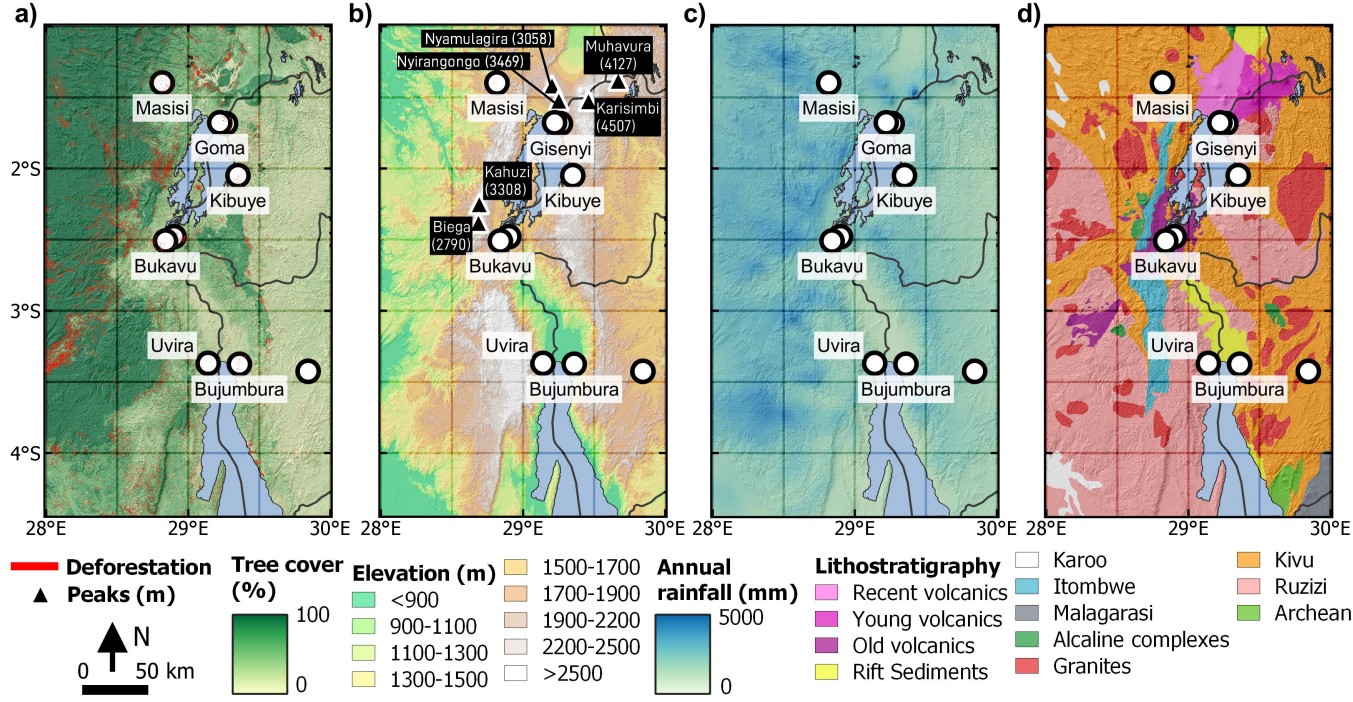

**Figure 2. Environmental characterization of the NTK Rift. a)** Tree cover for the year 2000 and deforestation from 2001-2017 (Hansen et al., 2013). **b)** Elevation and some renowned mountain peaks (height expressed in meter) (USGS, 2006). **c)** Average annual rainfall between 2005-2015 (Van de Walle et al., 2020). **d)** Lithostratigraphy (**Table 1**) (Laghmouch et al., 2018).

land that used to be forested in the past and 'deforested' land is the elapsed time since deforestation. Thus, the 'non-forest' land either underwent deforestation before the year 2000, or was never forest in the first place. 'Deforested' land experienced

deforestation over the last two decades.

### 3.1.2   Landscape rejuvenation

To distinguish the rejuvenated landscapes within the Rift shoulders from the surrounding landscapes (hereafter referred to as the relict landscapes), we use the spatial pattern of knickpoints retreating upstream towards the Rift shoulders, away from the active faults. Stationary knickpoints, here defined as knickpoints at a distance shorter than 1 km from a geological contact,

are considered unrelated to the rejuvenation process and removed from the analysis (Kirby and Whipple, 2012; Bennett et al., 2016). Two criteria are applied to identify the rejuvenated Rift: (i) the area must drain towards Lake Kivu or Lake Tanganyika, and (ii) the area must be located downstream of any non-stationary knickpoint, unless there is no knickpoint observed in the area. In the latter case, we assume the knickpoint reached the Rift shoulder and the landscape is completely rejuvenated.

We use the KNICKPOINTFINDER function in TopoToolbox to identify knickpoints. This function requires a drainage network

and tolerance value, reflecting the maximum expected error in the true river profiles (Schwanghart and Scherler, 2017). The

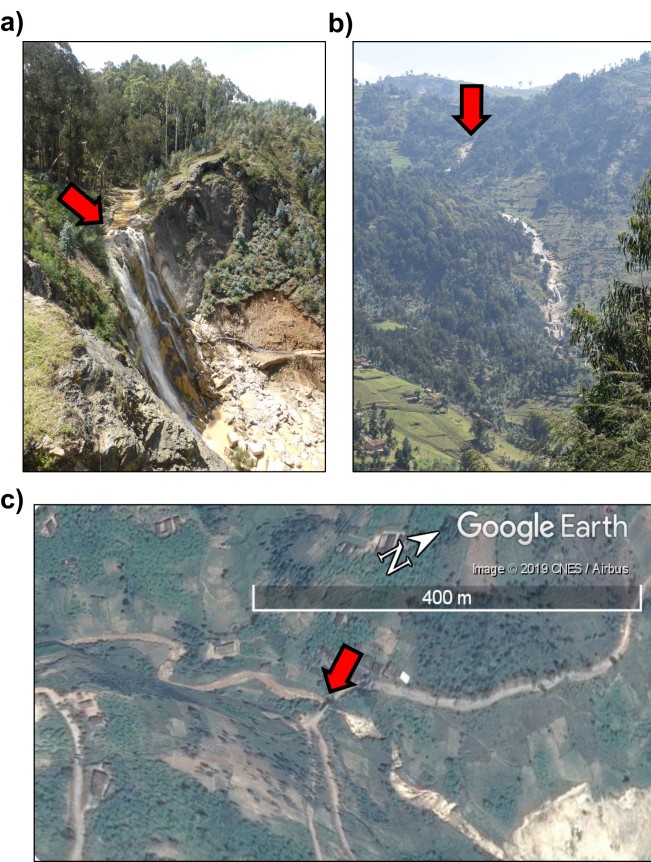

**Figure 3. Field-validated knickpoints in Rwanda.** The red arrow indicates the location of the knickpoint. **a)** East of Mabanze, Rwanda (29.469678° E, 2.047852° S). **b)** East of Kanama, Rwanda (29.391763° E, 1.705683° S). **c)** Southwest of Kibilira, Rwanda (29.61586° E, 1.980443° S; image ©2019 Google Earth).

drainage network for this purpose is modeled with the 1 arc second SRTM DEM data (USGS, 2006) and a threshold catchment area of $2 \times 10^6$ m$^2$. The tolerance value is used to distinguish knickpoints from discrepancies in the longitudinal river profile that are caused by errors in the DEM data. The tolerance value is calculated as the maximal difference between the 90[th] and 10[th] quantile of the smoothed river profiles (Schwanghart and Scherler, 2017), and subsequently lowered until the algorithm identifies the three knickpoints we validated in the field (**Fig. 3**).

### 3.1.3 Rainfall

Active rifting does not only trigger landscape rejuvenation but also impacts local rainfall patterns (Van de Walle et al., 2020). Depicker et al. (2020) showed a significant link between landslide occurrence and the frequency of extreme rainfall events in the NTK Rift. Moreover, field observations and local reports confirm that the majority of recent shallow landslides are rainfall-triggered (Monsieurs et al., 2018a; Depicker et al., 2020; Dewitte et al., 2021). To explore any relationship between

rejuvenation and rainfall, we analyze two metrics: the average annual rainfall and the number of times when accumulated rainfall was sufficiently large to trigger landsliding. As a proxy of the latter criterion, we use a 2 day, 15 mm threshold as it is a conservative estimation for global thresholds set by Guzzetti et al. (2008). Our intent is not to approximate an actual in-situ threshold, but rather to reflect spatial patterns in intense rainfall capable of triggering landslide events. For the comparison of rainfall patterns between rejuvenated and relict landscapes, we apply the non-parametric Mann-Whitney $U$ test, whereby each observation is the average metric (rainfall or threshold exceedance) in fifth order catchments. These units are derived from the 30 m DEM using a river catchment threshold of $10^5$ m$^2$.

The rainfall pattern is derived from a regional climate simulation with COSMO-CLM, a physical model, for the period 2005-2015 and using the ERA5 reanalysis product for the initial and boundary conditions of the atmosphere (Van de Walle et al., 2020; Hersbach et al., 2020). Due to in-situ data scarcity, evaluation of the simulated precipitation amounts is restricted to a comparison with a set of satellite products. Generally, the satellite data suggests lower amounts of rainfall compared to the simulated model output (Van de Walle et al., 2020), yet this was expected as satellite products tend to underestimate actual precipitation (Dinku et al., 2011; Monsieurs et al., 2018b). The final model output has a spatial resolution of 2.8 km and a temporal resolution of 1 hour (**Fig. 2c**). Note that the spatial resolution of the rainfall products might be too low to fully capture the impact of orographic controls and the local convective storm patterns (Monsieurs et al., 2018b).

| Age | Chronostratigraphy | Lithostratigraphy | Main lithological constitution |
|---|---|---|---|
| 10 Ka - present | Late Quaternary | Recent volcanics | Lava, tuff, and ash, deposited in the past decades and centuries, a result of eruptions of the Nyiragongo and Nyamulagira. |
| 2–1 Ma | Early Quaternary | Young volcanics | Relatively fresh basalts, deposited ± 2 Ma ago. |
| 12–6 Ma | Neogene | Old volcanics | Highly weathered basalt, deposited 11-4 Ma years ago. |
| 23 Ma - present | Late Cenozoic | Rift sediments | Sand along the lake or swamps more inland. |
| 360–201 Ma | Karoo | | Black shales, tillite, not metamorphosed. |
| 1000–540 Ma | Neoproterozoic | Itombwe | Black shales, tillite, silicified tillite, weakly methamorphosed. |
| | | Malagarasi | Black shales, tillite, silicified tillite, weakly methamorphosed. Presence of dolomites and volcanic rocks (basalts). |
| 820–720 Ma | | Alcaline complexes | Granitic rocks, intrusive volcanic rocks (rhyolite). |
| 1375–980 Ma | Mesoproterozoic | Granites | Two-mica and leucogranites. |
| 1600–1000 Ma | | Kivu | Pelites, quartzopelites, and quartzites at different degrees of weathering. Moderately metamorphosed. |
| 2500–1600 Ma | Paleoproterozoic | Ruzizi and ante-Ruzizi | Gneiss and micaschists, prone to chemical weathering, and quartzites, resistant to weathering. Strongly methamorphosed. |
| 4000–2500 Ma | Archaen | | Gneiss and micaschists, prone to chemical weathering, and quartzites, resistant to weathering. |

**Table 1. Lithostratigraphical units in the NTK Rift**, as presented by Depicker et al. (2020), and based on the work of Laghmouch et al. (2018).

### 3.1.4 Rock strength and threshold slopes

In order to account for the control of lithology on the hillslope response to uplift and incision (Schmidt and Montgomery, 1995; Korup, 2008; Korup and Weidinger, 2011; Bennett et al., 2016), we classify the 12 lithostratigraphical units present in the NTK Rift (**Fig. 2d** and **Table 1**) into major categories based upon the analysis of their threshold slope, a proxy for rock strength (Korup and Weidinger, 2011). Rock strength is a factor that must be taken into account when investigating landslide characteristics; equal slopes with different rock strength properties are expected to display different behavior in terms of landsliding and knickpoint retreat (Parker et al., 2016; Baynes et al., 2018; Campforts et al., 2020). We determine the rock strength by analyzing the dependency of the mean hillslope gradient, $S$, on the normalized steepness index, $k_{sn}$, averaged on a catchment scale (Safran et al., 2005; DiBiase et al., 2010; Bennett et al., 2016). We analyze first order catchments, whereby a drainage network was derived from the 1 arc second SRTM DEM data and a threshold catchment area that was set at $10^5$ m$^2$, i.e. large enough so that the smallest rivers visible in © Google Earth were detected. For each lithostratigraphical unit, we only consider watersheds where more than 50 % of the area is covered with the dedicated lithostratigraphy.

The $k_{sn}$ values of a river segment is a proxy for the river incision rate and is calculated using the following equation (Wobus et al., 2006):

$$k_{sn} = S_{chan} A^{\theta_{ref}}, \tag{1}$$

with $S_{chan}$ the local channel slope, $A$ the upstream catchment area, and $\theta_{ref}$ the reference concavity index, for which we assume a value of 0.45 (See e.g. DiBiase and Whipple, 2011).

Theory suggests a positive linear relationship between $S$ and $k_{sn}$ in catchments with relatively low river incision rates. For catchments with high river incision rates, an increase in $k_{sn}$ will not lead to further hillslope steepening, but to slope failure (DiBiase et al., 2010; Korup and Weidinger, 2011; Larsen and Montgomery, 2012), so that the $S$ becomes independent of the $k_{sn}$. To capture this non-linear dependency of average basin slope on channel steepness, we introduce a new empirical relationship to describe the response of $S$ to $k_{sn}$:

$$S = TA \left( 1 - \exp\left( - \frac{a}{TA} k_{sn} \right) \right), \tag{2}$$

where parameter $a$ is the slope of the curve at $k_{sn} = 0$. Thus, for low incision rates, $a$ approximates the slope of the linear relationship between $S$ and $k_{sn}$. Parameter *TA* is the slope angle to which the terrain converges for high $k_{sn}$ values. Hence, *TA* can be considered equivalent to the threshold slope. However, when there is a linear relationship for $S = f(k_{sn})$ in the entire $k_{sn}$ range (when the $R^2 > 0$ for a linear fit), we do not consider the threshold estimate reliable.

## 3.2 Quantifying shallow landslide erosion

### 3.2.1 Inventory

The assessment of shallow landslide erosion is based on a © Google Earth landslide inventory which is an update from the dataset presented by Depicker et al. (2020). Only recent landslides, for which we can estimate the time of occurrence,

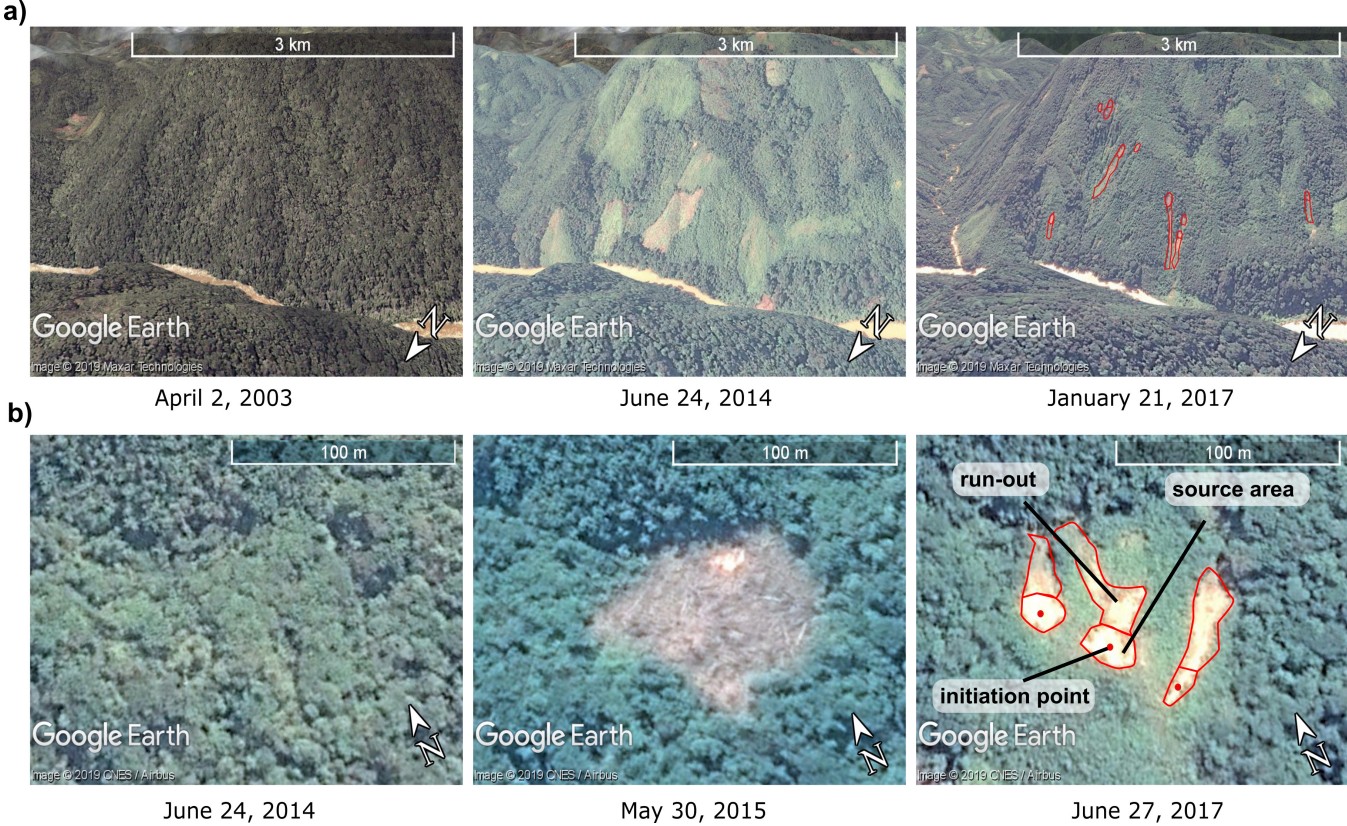

**Figure 4. Examples of deforestation followed by landsliding.** Left panels: prior to deforestation. Middle panels: after deforestation but prior to landsliding. Right panels: after landsliding. **a)** Landslide event north of Butezi, DRC (28.296984° E, 2.843201° S; image ©2019 Google Earth). **b)** Landslide event in Matale, DRC (28.360656° E, 2.645874° S; image ©2019 Google Earth).

are considered in our inventory. In other words, the moment of failure must be situated between the timing of two images. Moreover, since deforestation mainly affects the stability of the first few meters of the regolith (Sidle and Bogaard, 2016), we only consider shallow landsliding in this study. Deep-seated and bedrock landslides are excluded from the inventory. We apply a maximum depth of a couple of meters for landslides to be inventoried. We estimate the relative depth of the landslides (shallow or deep-seated) through *in situ* field observations and/or by visually analyzing the shape and size of the landslide scarp and deposits in © Google Earth imagery (Depicker et al., 2020). All images used in the analysis are of very-high spatial resolution, ranging from 30 to 60 cm. The images in © Google Earth are provided by either © DigitalGlobe or © CNES/© Airbus and they were captured between 2000 and 2019. Each landslide is manually assigned a polygon delineating the source area so that the total source area $LS_S$ can be calculated (m² km$^{-2}$ year$^{-1}$; Section 3.2.2). The $LS_S$ is the area over which regolith material has been removed by landsliding on an annual basis and serves as a proxy for shallow landslide erosion. To each landslide, we also manually assign a point of initiation used to calculate the landslide frequency $LS_F$ (#LS km$^{-2}$ year$^{-1}$; Section 3.2.2). In order to calculate the $LS_F$ as accurately as possible and avoid amalgamation, we differentiate between separate source areas

(Li et al., 2014; Marc et al., 2015; Roback et al., 2018). We also pay attention to not inventory landslides linked to mining and quarrying. Such sites were identified either during fieldwork, or in © Google Earth imagery through characteristics such as a gradual growth of the affected area over a time span of several years and the presence of mining infrastructure (road tracks, trucks, buildings, spoil tips) within the affected area.

The one-sided Mann-Whitney $U$ test is applied to statistically quantify any differences between different landslide populations (McDonald, 2014). Furthermore, we illustrate the potential differences between the landslide areas of rejuvenated and relict landscapes by comparing the frequency density of the landslide areas. The frequency density curves are fitted to the inverse $\Gamma$ distribution (Malamud et al., 2004).

### 3.2.2 Calculating landslide erosion rates from a biased © Google Earth inventory

Generally, the $LS_F$ is calculated as:

$$LS_F = \frac{n}{r\,A} \tag{3}$$

with $n$ the total number of shallow landslides, $A$ the total area (km$^2$), and $r$ the imagery range (years) in © Google Earth, i.e. the age difference between the oldest and youngest image. The imagery range is thus equal to the period of observation. However, the imagery range is highly variable throughout the study area due to differences in the availability of © Google Earth imagery (**Fig. 5a**). Since **Eq. (3)** is valid when $r$ is constant within our study area, we divide our study area in subareas $j$ that each have a constant imagery range $r^j$. The $LS_F$ in each subarea $LS_F^j$ is then:

$$LS_F^j = \frac{n^j}{r^j\,A^j}, \tag{4}$$

with $n^j$ the number of landslides in subarea $j$, $A^j$ the surface area of $j$, and $r^j$ the constant imagery range in $j$. To calculate the frequency for the entire study area, the frequencies $LS_F^j$ are averaged out using weights proportional to their corresponding area $A^j$:

$$LS_F = \sum_{j=1}^{N} \frac{A^j}{A} LS_F^j, \tag{5}$$

with $N$ the number of subareas $j$. Substituting **Eq. (4)**, **Eq. (5)** becomes:

$$LS_F = \frac{1}{A} \sum_{j=1}^{N} \frac{n^j}{r^j}. \tag{6}$$

Hence, we do not require the size $A^j$ of each subarea $j$ for the calculation of the total $LS_F$. Instead of aggregating the $LS_F$ over all subareas, we can aggregate the $LS_F$ over the individual landslides. **Equation (6)** then becomes:

$$LS_F = \frac{1}{A} \sum_{i=1}^{n} \frac{1}{r^i}, \tag{7}$$

with $r^i$ the time range observed in landslide $i$. The landslide inventory is expected to be biased due to spatial differences in the imagery density $d$ (**Fig. 5b**), defined as the total number of available images at each location, as vegetation regrowth might

erase the spectral signature of landslides before they are captured in imagery. Hence, we expect to detect more landslides in areas with higher imagery density. To compensate for this bias, we assume that the probability of identifying a landslide in a certain region increases linearly with imagery density in that specific region. **Equation (7)** then becomes:

$$LS_F = \frac{1}{A} \sum_{i=1}^{n} \frac{1}{r^i d^i}, \tag{8}$$

with $d^i$ the imagery density observed at the location of landslide $i$. Note that there can be a saturation of the information provided by the imagery: when the imagery density is high, the availability of one extra image will have no to little effect on the observed number of landslides. We validate our assumptions of linearity and saturation by visually assessing the dependency of landslide density (# landslides km$^{-2}$) on imagery density. If the assumption of linearity does not hold, we have to apply a non-linear transformation on the $d^i$ values. If saturation is problematic to our inventory, we have to set a maximum value for $d^i$.

Deriving the $LS_S$ equations is analogous to deriving the ones for the $LS_F$. We only have to slightly modify **Eq. (7)** and **Eq. (8)**:

$$LS_S = \frac{1}{A} \sum_{i=1}^{n} \frac{a_{src}^i}{r^i}, \tag{9}$$

$$LS_S = \frac{1}{A} \sum_{i=1}^{n} \frac{a_{src}^i}{r^i d^i}, \tag{10}$$

whereby $a_{src}^i$ is the source area of landslide $i$. Note that the calculation of $LS_S$ will be less accurate than for the $LS_F$ due to biases in the delineation of the landslide source area. These biases are caused by the time lag between the landslide occurrence and the landslide detection in © Google Earth, whereby part of the source area might already have recovered. To avoid biases linked to the interpretation of the source area, all landslides were delineated by the same person. In order to statistically verify a difference in landslide activity between regions (for example rejuvenated versus relict landscapes), we use the one-sided non-parametric Mann-Whitney $U$ test to compare the different landslide activity measures in fifth order water catchments (calculated with **Eq. (8)** and **Eq. (10)** to compensate for imagery density differences).

### 3.2.3 Impact of slope on landslide erosion

In order to assess the impact of slope steepness on the $LS_S$ (a proxy for landslide erosion), we first reclassify the slope values between 0-50° into 10 classes of equal width, and subsequently apply **Eq. (10)** to each slope class and the landslides therein. Similarly, to assess the impact of slope steepness on $LS_F$, we apply **Eq. (8)** to each slope class and its landslides. Furthermore, we estimate the degree to which our $LS_S$ and $LS_F$ calculations are affected by outliers and/or extreme landslide events. First, we divide the study area in 50 east-west bands of equal width. Second, we calculate the $LS_S$ and $LS_F$ for each slope class 50 times, each time leaving out the slope and landslide data for a single east-west band. In other words, for each run we slightly perturbate the landslide inventory.

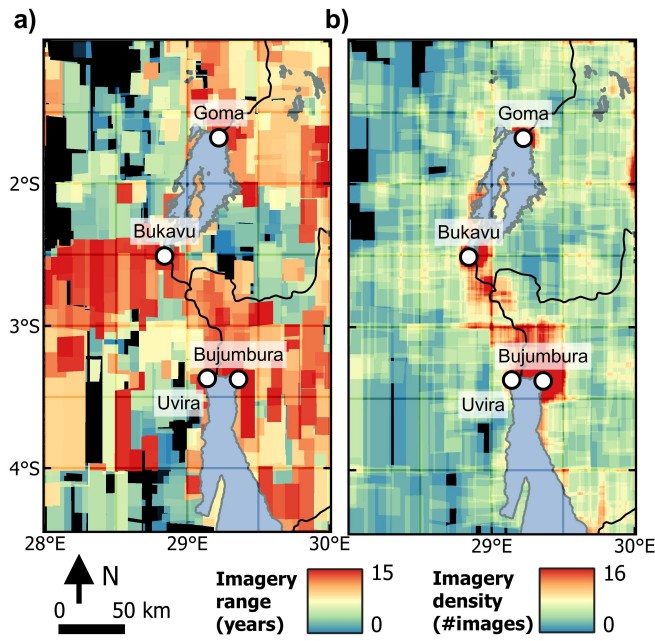

**Figure 5. Visualization of the imagery bias in © Google Earth.**, prior to May 4, 2019. **a)** Imagery range. **b)** Imagery density. The range and density were calculated by manually identifying 932 imagery footprints. The highest imagery density is available for the major cities in the study area (Goma,Bukavu, and Bujumbura), whereas the northwest and southwest regions have fewer observations. For some areas (in black) there is no available image.

### 3.2.4 Linking forest cover and deforestation to landslide erosion

In order to link forest dynamics to landslide erosion, we must distinguish between landslides that followed deforestation (**Fig. 4**) and landslides that caused deforestation. To identify the correct causality, we reconstructed the timeline of every landslide that occurred on deforested land (**Fig. 6b**). Landslides following deforestation are defined as those that happened within the post-deforestation time range, being the period between the first image after the year of deforestation and the most recent image.

Determining the $LS_S$ in function of the time elapsed since deforestation ($t_{def}$) is necessary to characterize the post-deforestation landslide wave. Because $t_{def}$ is temporally dynamic this analysis requires two components: i) the total area $A_{t_{def}}$ in which we can observe land that was deforested $t_{def}$ years ago, and ii) the total affected area of landslides that happened $t_{def}$ years after deforestation. The first component, $A_{t_{def}}$, entails all areas where the sum of $t_{def}$ and the year of deforestation lies in the time range between the age of the oldest and newest post-deforestation image in © Google Earth. For the second component, we only include landslides for which the time between deforestation and landsliding ($t_{def \to LS}$) is equal to $t_{def}$.

There is a considerable degree of uncertainty associated to $t_{def \to LS}$ since we do not know the exact timing of the landslides (the occurrence is situated between the capture times of the image where it was initially observed and the preceding image).

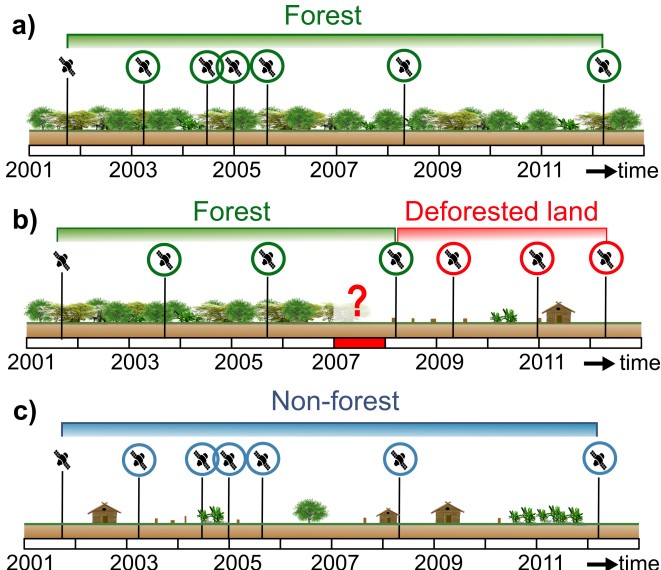

**Figure 6. Schematic overview of the three considered forest cover scenarios in © Google Earth.** The satellite icons signal the availability of a © Google Earth image and the colored circles indicate whether we can potentially observe recent landslides in the concerned image. **a)** Forest scenario: each landslide observed in these areas is linked to forest cover. **b)** Deforestation scenario: only landslides observed starting from the second © Google Earth image after the year of deforestation are considered to be linked to deforestation (in other words, we can only observe deforestation-induced landslides in imagery that is encircled in red on the figure). Hence, in this illustrated example, we cannot attribute landslides from the 2008 imagery to deforestation, as we cannot be sure that these landslides happened before or after the 2007 deforestation. Note that we do not know the exact moment of deforestation, only the year (indicated with the red bar) is reported. **c)** Non-forest scenario: every landslide observed in these regions is linked to non-forest.

Similarly, we know the year of deforestation but not the exact date. To assess the uncertainty on the timing of deforestation and landslide occurrence, we calculate the $LS_S$ 100 times, each time sampling a new $t_{def \rightarrow LS}$ for each landslide. For each sample of $t_{def \rightarrow LS}$, we make two assumptions: i) the exact moment of deforestation (the lower limit of $t_{def \rightarrow LS}$) is assumed to be distributed uniformly in the reported deforestation year. ii) The timing of landslide occurrence (the upper limit of $t_{def \rightarrow LS}$), is assumed to be distributed uniformly between the capture times of the image where it was initially observed and the preceding image.

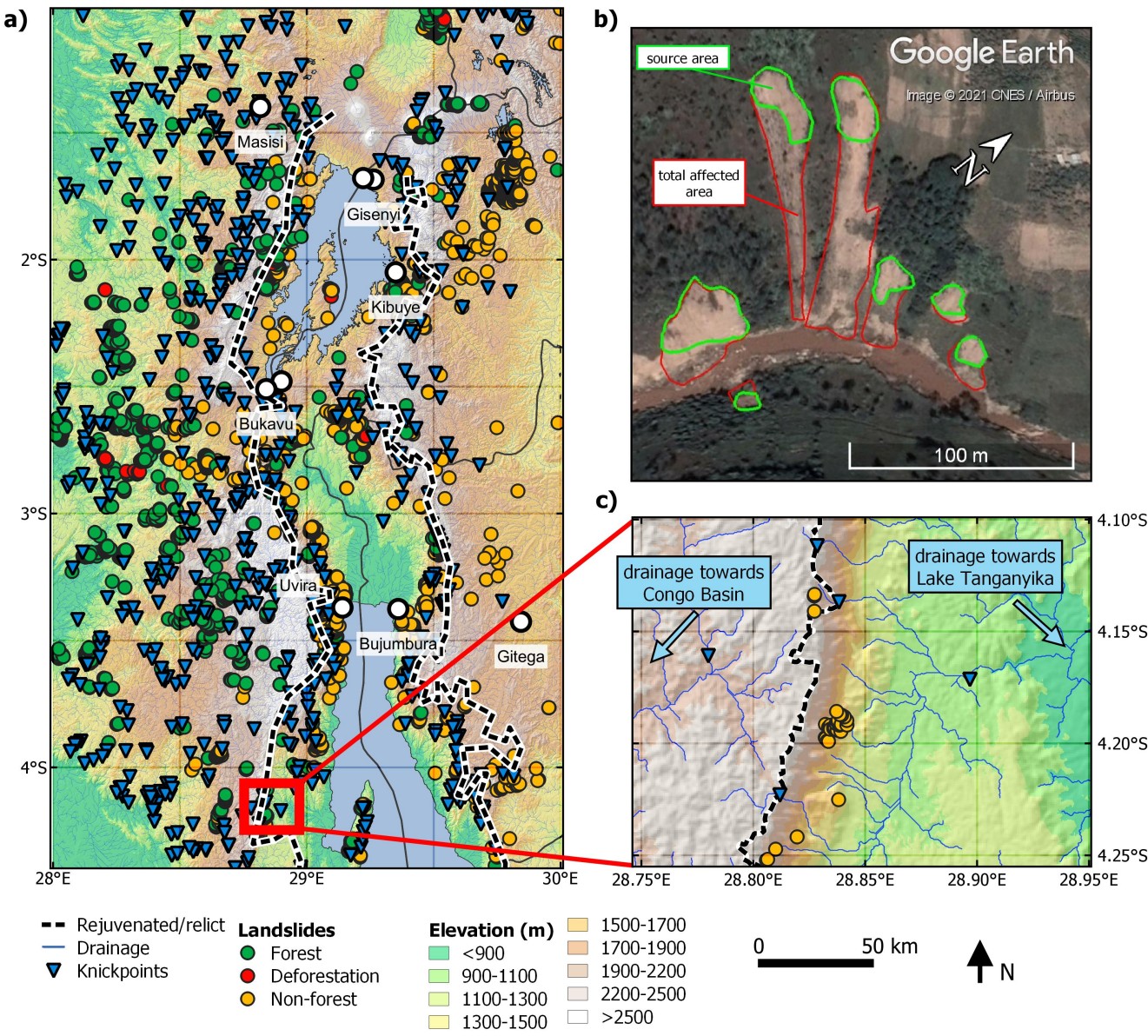

**Figure 7. Landslide and knickpoint inventory for the NTK Rift. a)** We identified 7,994 shallow recent landslides that occurred either in forest, non-forest, or after deforestation, and 673 non-stationary knickpoints. These knickpoints were used to separate the rejuvenated landscapes between the Rift shoulders from the surrounding relict landscapes (black-and-white line). **b)** Example of shallow landslides in Rwanda (29.7909 °E, -1.7151 °S) and the delineation of their total area (red) and source area (green). c) Example of the Rift shoulder west of Lake Tanganyika. The method for delineating the rejuvenated landscapes is specified in Section 3.1.2.

# 4 Results

## 4.1 Regional controls on landslide erosion

We identified 673 non-stationary knickpoints using a tolerance value of 100 m. These knickpoints were used to demarcate the rejuvenated landscapes inside the Rift (**Fig. 7a**). The rejuvenated landscapes encompass 15,526 km$^2$, or 18 % of the entire study area.

The average annual rainfall in the rejuvenated landscapes is significantly lower than in the relict landscapes (1,905 mm/year versus 2,297 mm/year, p<0.01). Similarly, we find that within the rejuvenated landscapes, the 2 day, 15 mm threshold is exceeded less often compared to the relict landscapes (17 % difference, p<0.01), indicating that intense (potentially landslide-triggering) rainfall events occur less frequently.

Based on the analysis of 234,840 first-order catchments, we identify three major lithological categories (**Fig. 8**). Category I comprises units that do not display clear threshold angles. These are lithostratigraphies of relatively young age such as recent and young volcanic basalts and lake and river sediments (all <23 Ma) except for the Malagarasi rock formations. The latter formations are of old age (1000-540 Ma) and cover only a small area in the southeast of the study area. The lack of threshold landscapes in Category I could be related to the relatively short duration of exposure to weathering for these rocks. Category II, consisting of old volcanic basalts and Karoo lithostratigraphy (both younger than 210 Ma), has threshold slopes of roughly 17°. Rocks of Category III, with observed threshold slopes ranging between 24-28°, are generally of older age (>540 Ma) and display a high resistance to slope failure. The lithostratigraphy of Category III includes the following formations: Itombwe, Alcaline complexes, Granites, Kivu, Ruzizi, and Archaen.

## 4.2 Shallow landslide erosion in the NTK Rift: impacts of deforestation and rejuvenation

We inventoried 7,944 recent shallow landslides (**Fig. 7a**). Following the classification of Hungr et al. (2014), the observed landslides were mostly debris slides, caused by the sliding of regolith on a planar surface parallel to the ground. These debris slides, once initiated, often transform into avalanches, characterized by the flow of (at least partially) saturated debris on a steep slope. Another commonly observed landslide type was the debris/mud flow, defined as the rapid flow of saturated debris in a steep channel. In total, we found 873 landslides in deforested land, yet for only 378 of those landslides we could be certain that they were preceded by deforestation (Section 3.2.4). Furthermore, 3,155 landslides were associated with forest, and 4,411 with non-forest. Rocks of Category I and II combined contained only 344 instances, hampering a robust analysis. We therefore focus our further analysis on the 7,600 landslides in areas with rocks of Category III.

The number of observed landslides in © Google Earth appears to increase linearly with the available imagery up to a density of 12 images (**Fig. 9a**). The proportion of the study area with a higher density than 12 images is negligible (1.5 %) and contains merely 1 % of all landslides in the inventory. Hence, the assumption that landslide density is linearly dependent on imagery density is valid within our study area and we take no measures to correct for saturation (Section 3.2.2). The annual extent of imagery made available in © Google Earth increases with time, especially after 2010 (**Fig. 9b**).

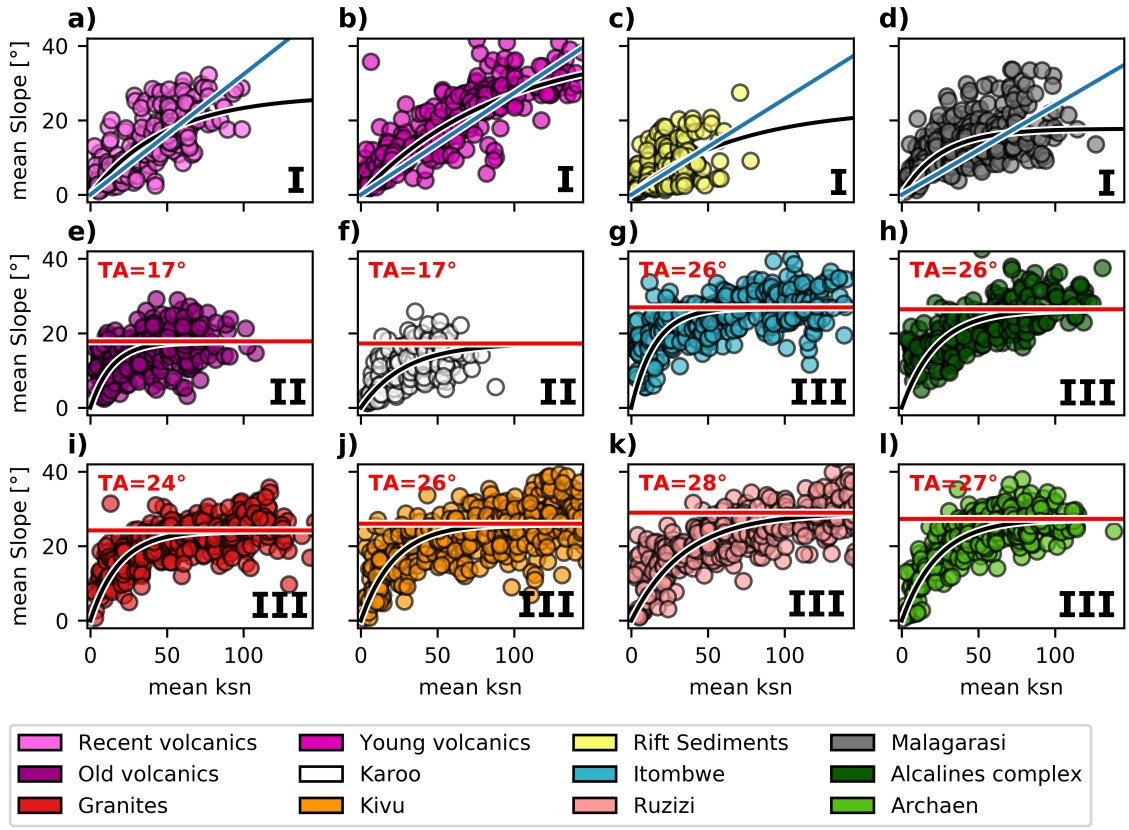

**Figure 8. Threshold slope analysis for the different lithostratigraphical units in the NTK Rift.** Each point represents a first order river catchment over which we averaged the slope gradient $S$ and normalized river steepness index $k_{sn}$. The black curves represent the $S = f(k_{sn})$ relationship fitted to **Eq.** (2). **a)→d) Category I** Young lithostratigraphy for which no clear threshold angle is observed. **e)→f) Category II** Young lithostratigraphy with a low threshold angle of ca. $17°$. **g)→l) Category III** Older rocks with higher observed threshold slopes of $24-28°$.

After accounting for differences in imagery density, the overall $LS_S$ in rejuvenated landscapes, which is a proxy for shallow landslide erosion, is roughly 40 % higher than in relict landscapes (p=0.034, **Fig. 10a**). The difference becomes even larger when looking at the $LS_F$ (160 %, p=0.014, **Fig. 10b**), which implies that landslides are on average smaller in rejuvenated landscapes. This difference in landslide size between rejuvenated and relict landscapes is confirmed in all three land cover types: forests (114 versus 308 m$^2$, p<0.01, **Fig. 11a**), non-forests (111 versus 138 m$^2$, p<0.01, **Fig. 11b**), and deforested land (94 versus 239 m$^2$, p<0.01). Similar to the rejuvenation status, forest cover also influences the landslide size. The average source area for forests (223 m$^2$) decreases non-significantly after deforestation (165 m$^2$, p=0.06). In non-forest lands, the landslide size (126 m$^2$) is significantly smaller than in recently deforested lands (p<0.01).

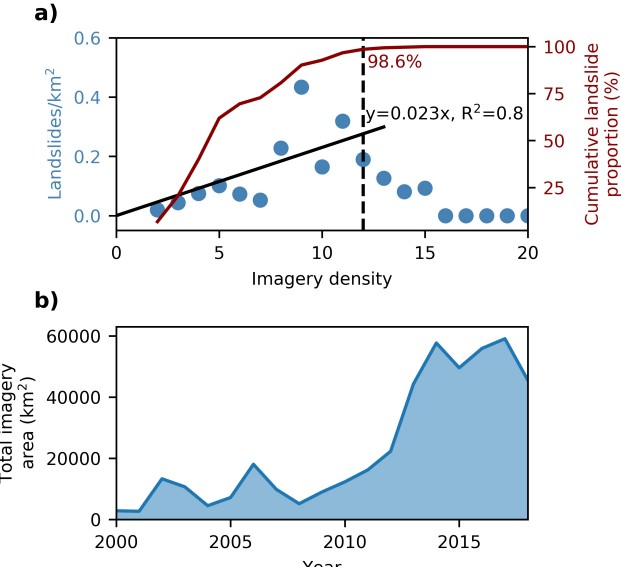

**Figure 9. The impact of imagery density (the number of available images in © Google Earth) on the number of observed landslides.**
We only show the results for the rocks of Category III (Section 4.1). Therefore, major cities that are characterized by a high imagery density like Bukavu, Bujumbura, and Goma are excluded from this figure. **a)** Impact of the imagery density on the number of observed landslides. The number of landslides seems to increase linearly with imagery density up to 12 images. The cumulative landslide proportion for a certain value shows the % of the landslide inventory contained in areas with an imagery density equal or lower than that value. **b)** The evolution of imagery availability between 2000 and 2018.

The $LS_S$ and $LS_F$ increase with slope gradient (**Fig. 12a-b,d-e**). A decrease is observed for forested slopes >45°, which could be linked to limitations on regolith formation, whereby weathering and sediment deposition are outpaced by erosion (Montgomery, 2001; Dykes, 2002; Prancevic et al., 2020). When comparing slopes of equal steepness, we observe that the
$LS_S$ is generally higher in relict landscapes than in rejuvenated landscapes (**Fig. 12c**). Nevertheless, the overall $LS_S$ is higher in rejuvenated landscapes, because the overall predominance of steeper relief (**Fig. 12g-h**) compensates for the fact that, comparing similarly angled individual slopes in rejuvenated and relict zones, rejuvenated slopes are shown to have a lower or equal rate of shallow landslide erosion (**Fig. 12c**).

Recently deforested slopes are up to eight times more sensitive to shallow landsliding compared to forested slopes (**Fig. 12a-**
**b**). The deforestation effect lasts approximately 15 years (**Fig. 13**). However, deforestation increases $LS_S$ much more in relict landscapes compared to rejuvenated areas (**Fig. 12c**). The $LS_S$ in the 'non-forested' areas (blue lines in **Fig. 12**) corresponds to the situation that prevails once the deforestation-induced landslide wave has passed. In this situation, the $LS_S$ drops back to a level similar to that observed under forest (green line) (**Fig. 12a-b**).

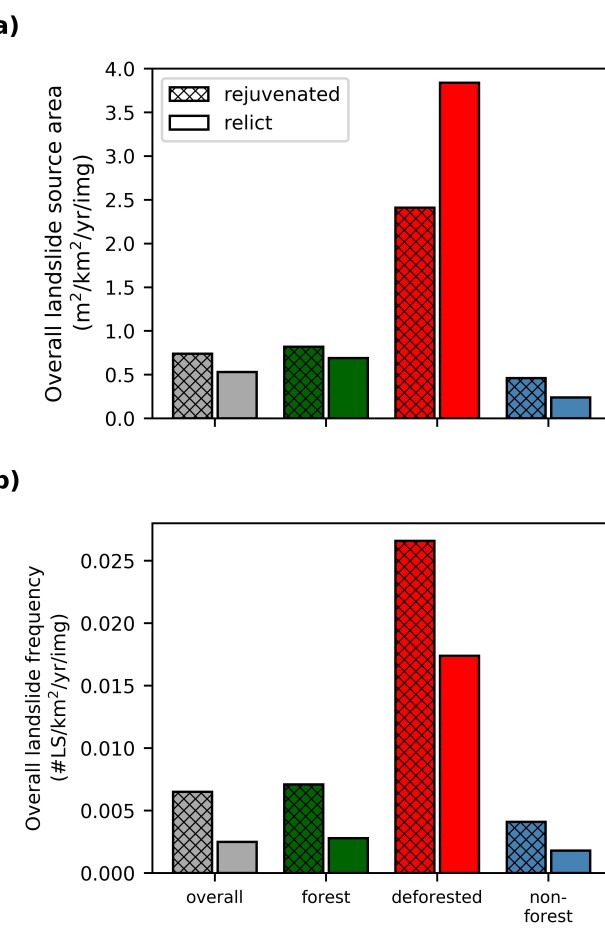

**Figure 10. Total landslide activity in the NTK Rift**, adjusted for imagery density. **a)** Total landslide source area ($LS_S$), a proxy for landslide erosion. **b)** Landslide frequency ($LS_F$).

## 5   Discussion

### 5.1   Interactions between deforestation, rejuvenation, and landslide erosion

While the landslide erosion rate (approximated by the $LS_S$) is higher in rejuvenated landscapes due to a steeper relief, the relative effect of slope steepness on landslide erosion appears to be weaker in rejuvenated landscapes: we found that steep ($>35°$) forested slopes display higher shallow landslide erosion rates in relict landscapes than in rejuvenated landscapes (**Fig. 12c**). We evaluate three mechanisms that could explain this difference: seismic activity, regolith availability, and climate.

Seismic activity is a first factor that could explain why slope has a different impact on landslide erosion in rejuvenated and relict landscapes. Generally, there is more and stronger seismic activity within the rejuvenated landscapes (Delvaux et al., 2017). We hypothesize that the higher seismic activity would result in elevated landslide erosion rates on longer timescales

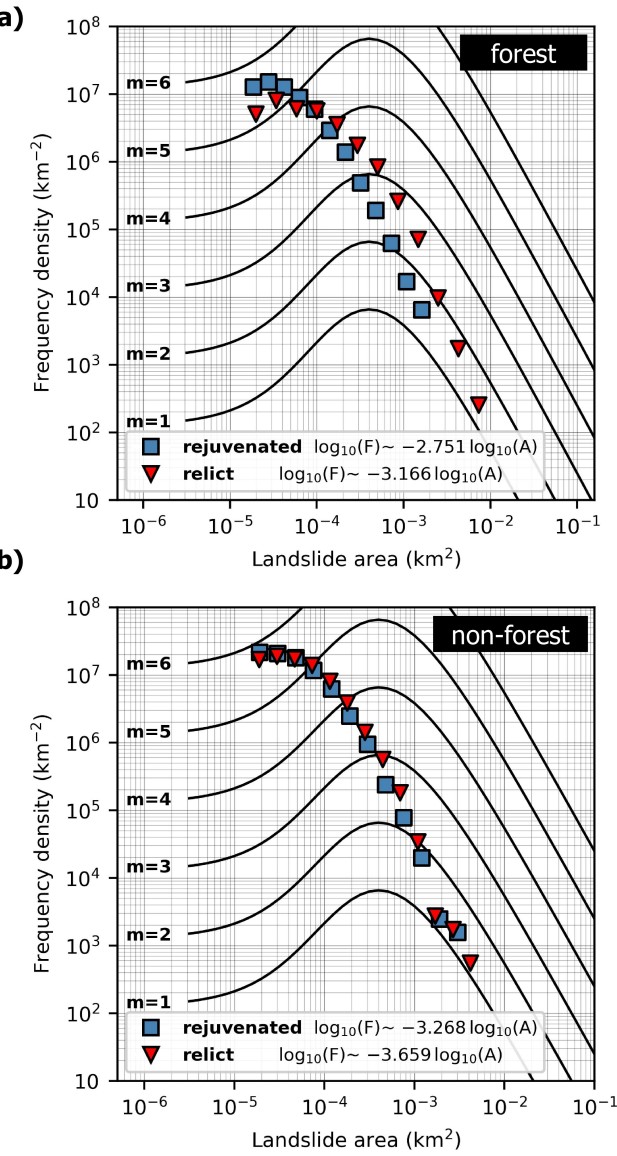

**Figure 11. Frequency density in function of the landslide source area. a)** The area frequency density of shallow landslides in forest, separated for rejuvenated and relict landscapes. **b)** The area frequency density of shallow landslides in non-forest, separated for rejuvenated and relict landscapes. There were not enough landslide observations in deforested land to fit to their area frequency density to the inverse Γ distribution. The general frequency density distributions for inventories of different magnitudes (the black lines on the curves) are derived from Malamud et al. (2004). Note that the frequency is skewed towards smaller landslide sizes due to the omission of deep-seated (and generally larger) landslides from our inventory.

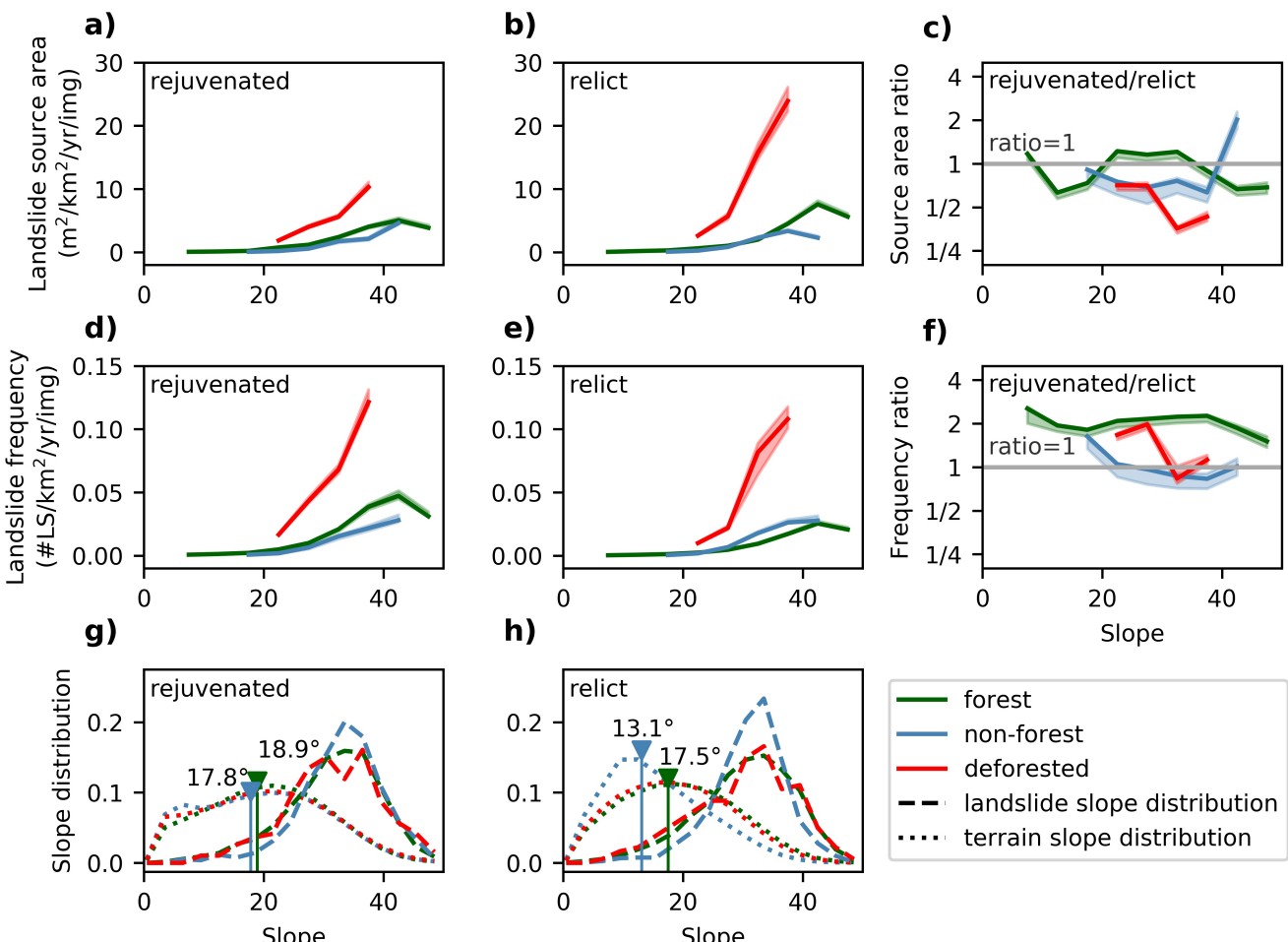

**Figure 12. The effect of slope steepness and rejuvenation on landslide activity**, corrected for imagery density. We only show results for slope classes in which we observed more than 20 landslides. **a)→c)** Landslide source area ($LS_S$) in function of slope. **d)→f)** Landslide frequency ($LS_F$) in function of slope. **g)→h)** Slope distribution for the terrain and landslides in the rejuvenated and relict landscapes. The blue and green arrows indicate the median slope in non-forest and forest landscapes. The slopes in rejuvenated landscapes are clearly steeper, both in forest and non-forest land.

due to the occurrence of major landslide events triggered by large earthquakes (Delvaux and Barth, 2010; Marc et al., 2015). However, in our observed period, chances of earthquake-triggered landsliding were very limited (Dewitte et al., 2021). The lack of such observations suggests that our window of observation was too short to capture earthquakes that were large enough to trigger landsliding. Over the long term, the contribution of earthquake-induced landsliding to regolith mobilization in the rejuvenated landscapes may nevertheless be important. Earthquakes fracture and weaken the hillslope material and hence reduce the minimum critical area for landslide initiation (Delvaux et al., 2017; Milledge et al., 2014; Vanmaercke et al.,

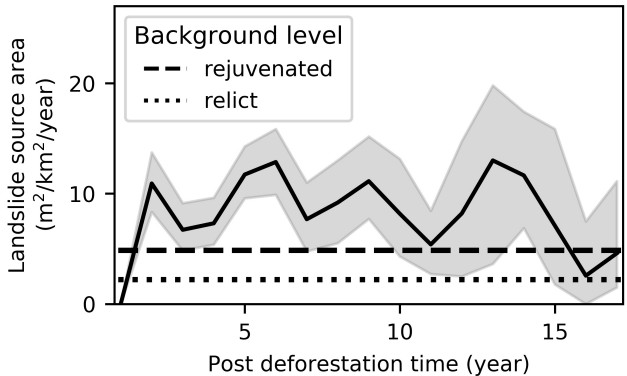

**Figure 13. Deforestation-induced landslide wave.** Total landslide source area ($LS_S$, m² km⁻² year⁻¹) in function of time elapsed since deforestation, based on the analysis of 374 post-deforestation landslides in rocks of category III (Section 4.1). The grey area is the 90 % confidence interval, derived from 100 iterations of $LS_S$ calculations (Section 3.2.4). The dashed and dotted line represent the overall erosion rates in rejuvenated and relict landscapes. There are not enough observations to make two separate consistent plots for rejuvenated and relict landscapes (**Fig. A1**)

2017). As such, seismic activity may also contribute to a smaller average landslide size in rejuvenated landscapes. Moreover,
a previous study in the NTK Rift established an indirect link between spatial patterns of seismic activity (approximated by a modelled PGA product by Delvaux et al. (2017)) and the spatial pattern of the landslide occurrence, though this study did not differentiate between deep-seated and shallow landsliding (Depicker et al., 2020).

A second factor potentially contributing to the difference in slope impact on erosion rates in rejuvenated and relict landscapes (**Fig. 12c**) is that the regolith mantle on rejuvenated slopes is expected to be thinner and less continuous due to the drier climate,
the younger age of the landscape, the continuous adaptation to river incision, and sporadic earthquake-triggered landslide events (Schoenbohm et al., 2004; Egholm et al., 2013; Marc et al., 2015; Braun et al., 2016), thereby inducing a supply-limited landsliding regime. This can also partly explain why the rejuvenated landscapes have more, but smaller landslides in comparison to relict landscapes, as the size of the shallow landslides is constraint by regolith availability (Prancevic et al., 2020). However, we do not have direct evidence supporting this hypothesis: the collection of field data on regolith thickness
is hampered by limited access to the field, especially in the eastern DRC. Alternatively, regolith depth could be derived from landslide scars observed on a high-resolution DEM, but such product is currently not available.

A difference in the frequency of landslide-triggering rainfall events could be a third explanation for the lower impact of slope steepness on landslide erosion in rejuvenated landscapes. Based on the global rainfall threshold proposed by Guzzetti et al. (2008), we observe that the rainfall threshold for landsliding is exceeded more often in relict landscapes. However, due to
330 these differences in rainfall, we would not only expect a stronger response of erosion rate to slope steepness in relict landscapes (**Fig. 12c**), but also a higher $LS_F$. The latter is not the case: slope steepness appears to have a lower effect on $LS_F$ in relict landscapes than in rejuvenated landscapes (**Fig. 12f**). This discrepancy between erosion and frequency could be linked to

two factors: differences in regolith thickness which allow for larger landslides in the relict landscapes, and seismic fracturing allowing for smaller (and more) landslides in the rejuvenated landscapes.

Deforestation drastically increases the landslide frequency and landslide erosion rate. The observed landslide erosion and frequency increases two- to eight-fold after deforestation (**Fig. 12a-b**), which is the same order of magnitude as what has been reported in literature for other regions (Jakob, 2000; Guthrie, 2002; Glade, 2003). The effect of deforestation on landslide erosion and frequency is temporary, lasting approximately 15 years (**Fig. 13**; Sidle et al., 2006; Sidle and Bogaard, 2016). After the wave has passed, landslide erosion rates appear to decrease to a level similar to the average erosion rate in the region

(**Fig. 13**). However, a longer period of observation would be useful to confirm if this effect persists after 15 years.

    We find that the landslide erosion response to deforestation and in function of slope is much more pronounced in relict landscapes than in rejuvenated landscapes (**Fig. 12c**). This observation may be linked to the drier climate and higher seismic activity in the rejuvenated landscape: there are less landslide triggering rainfall events and earthquakes may induce a higher average landslide frequency thereby removing sensitive pockets of regolith on a semi-regular basis. Differences in regolith

availability can be invoked to explain the different response of these landscapes to deforestation. Assuming that rejuvenated areas are indeed devoid of regolith (in comparison to relict areas), one may expect that deforestation will lead to a less important response in rejuvenated areas, simply because the stock of material that can be mobilized through landsliding is smaller.

    The landslide erosion rates in non-forest land are much lower than in deforested areas and, in fact, similar to or lower than what has been observed in forests (**Fig. 12a-b**). Thus, even when there is no regrowth of forest vegetation, the landslide erosion

rate returns to normal levels some time after deforestation. A possible hypothesis that might explain this result is that once the effect of deforestation on landsliding has worn out, the regolith mantle is protected as efficiently by forest cover as by grassland and crops, despite the presence of human practices such as terracing that could promote landsliding (Sidle and Ochiai, 2006). However, this is extremely unlikely given the much smaller rooting depths of both grasses and crops (Holdo et al., 2018). A more probable explanation for the fading-out of the deforestation-induced landslide wave is therefore that the landslide

frequency and erosion rate return to lower levels once the most landslide-sensitive regolith pockets have been removed. For those slopes that are stripped of their regolith mantle after deforestation, the rainfall threshold for slope failure is temporarily increased and it may take thousands of years to redevelop a regolith depth that matches the pre-failure conditions (Dykes, 2002; Hufschmidt and Crozier, 2008; Parker et al., 2016). Additionally, depending on the properties of the landslide deposits (e.g. fine-grained or rock debris), slopes might also experience depositional hardening due to an increase in bulk density and

cohesion of the slope material (Crozier and Preston, 1999; Brooks et al., 2002), yet field data is required to test this hypothesis.

    Despite the fact that equal slopes in non-forest and forest land display similar landslide erosion rates, the average source area is significantly smaller in non-forest landscapes. The smaller size is likely due to the absence of trees and the associated lower overall root cohesion. Regolith with a lower root cohesion exhibits a smaller minimum critical area needed to initiate landsliding (Milledge et al., 2014; Sidle and Bogaard, 2016). Hence, landslides in non-forest and forest land have different size

characteristics (**Fig. 11**), but the total erosion rate in function of slope remains similar (**Fig. 12**).

## 5.2 A new approach for calculating landslides erosion rates?

Using **Eq. (9)** which deals with the biases in the © Google Earth imagery range, we obtain an overall $LS_S$ of 4.86 m$^2$ km$^{-2}$ year$^{-1}$ in rejuvenated landscapes. Using the volume~source area relationships presented by Larsen et al. (2010) for soil landslides in Uganda, we obtain a rough estimate of the landslide volumes. As such, we find that the $LS_S$ in rejuvenated landscapes corresponds to an erosion rate of ca. 0.006 mm year$^{-1}$. This rate can be compared to the regional uplift rates to estimate the importance of shallow landsliding in the overall evolution of the NTK Rift. There are no accurate estimates of the uplift rates in the study area, but the maximal estimation in the Rwenzori Mountains, a particularly tectonically active region located 150 km North of our study area, is 2 mm year$^{-1}$ (Kaufmann et al., 2016). If we consider similar rates in the NTK Rift, shallow landslide erosion compensates merely 0.3 % of the uplift in the rejuvenated landscapes, assuming a steady state between uplift and denudation. Based on a global relationship between mean local relief and erosion rate, formulated by Montgomery and Brandon (2002), we obtain a more conservative value of 0.6 mm year$^{-1}$ for the average erosion rate in landscapes with a similar mean local relief as the rejuvenated landscapes in the NTK Rift (ca. 1,300 m). In this scenario, shallow landslide erosion accounts for 1.0 % of the total erosion. Both the upper and lower estimate suggest that, while shallow landslides are highly visible in the landscapes we studied, their geomorphic effect is somewhat limited. However, it must be noted that the estimated erosion rate due to shallow landsliding is most likely an underestimation. First, we did not observe earthquake-induced landslide events, which are rare but may lead to catastrophic landslide erosion (Marc et al., 2015; Dewitte et al., 2021). Second, the landslide inventory used to calculate the erosion rate is incomplete due to limitations in © Google Earth coverage. Furthermore, we focused on shallow landsliding but other processes such as deep-seated landsliding also contribute significantly to erosion (Depicker et al., 2020; Dewitte et al., 2021). Nevertheless, it is to be expected that overall erosion rates are lower than uplift rates: this is the basic explanation as to why mountainous topography is formed.

## 6 Conclusions

We studied shallow landsliding along the NTK rift in order to understand how the interplay of landscape rejuvenation and deforestation affects landslide erosion rates. Rejuvenated landscapes display a higher shallow landslide erosion rate than relict landscapes. Contrarily, the relative effect of slope steepness on landslide erosion rates is smaller in rejuvenated landscapes. These two seemingly contradicting results are reconciled by the observations that erosion generally increases with slope gradient, and that the average slope is much steeper in the rejuvenated landscapes. The lower impact of slope steepness on landslide erosion in the rejuvenated landscapes could be the result of three factors: the omission of earthquake-induced landslide events in our inventory, a thinner regolith mantle, and a drier climate. The hypothesis is consistent with our observations that deforestation initiates a much larger landslide peak in relict landscapes and that landslides are, on average, much smaller in rejuvenated landscapes. Thus, the response of a landscape to deforestation does not only depend on local topography and climate but also on the geomorphic status of the landscape. Understanding this differential response is also important to assess the risk for the local population. Our study shows that such understanding is only possible if (i) inventory biases linked to © Google Earth

imagery are properly eliminated, (ii) landscape status (rejuvenated versus relict) is accounted for, and (iii) a sufficiently long time frame is considered to capture the transient nature of the deforestation-induced landslide wave.

*Author contributions.*  **Arthur Depicker** was responsible for the compilation of the inventory data, the conceptualization of the paper storyline, the development and execution of the statistical analyses, the conduction of fieldwork, and the writing of the manuscript. **Gerard Govers** was involved in conceptualising the paper storyline, shaping the discussion, writing of the manuscript, and obtaining funding for this work. **Liesbet Jacobs** helped to fine-tune the methodology and statistical analysis, to conceptualize the paper storyline, and to write the manuscript. **Benjamin Campforts** provided the know-how to calculate drainage networks, knickpoint locations, and watershed statistics in

the TopoToolbox. He contributed to the paper storyline and writing of the manuscript. **Judith Uwihirwe** was a key figure for the completion of fieldwork in Rwanda that lead to the identification of knickpoints and helped in improving our inventory. She provided feedback for the manuscript and helped to better understand landslide processes in the study area. **Olivier Dewitte** was involved in compiling the inventory, conducting fieldwork, conceptualizing the paper storyline, shaping the discussion, writing the manuscript, and obtaining funding for this work.

*Competing interests.*  The authors declare that they have no conflict of interest.

*Acknowledgements.*  This study was supported by the Belgium Science Policy (BELSPO) through the PAStECA project (BR/165/A3/PASTECA) entitled 'Historical Aerial Photographs and Archives to Assess Environmental Changes in Central Africa' (http://pasteca.africamuseum.be/). We would like to thank Jonas Van de Walle for the provision of the rainfall dataset used in this work.

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

**Appendix A: Deforestation wave in rejuvenated and relict landscapes**

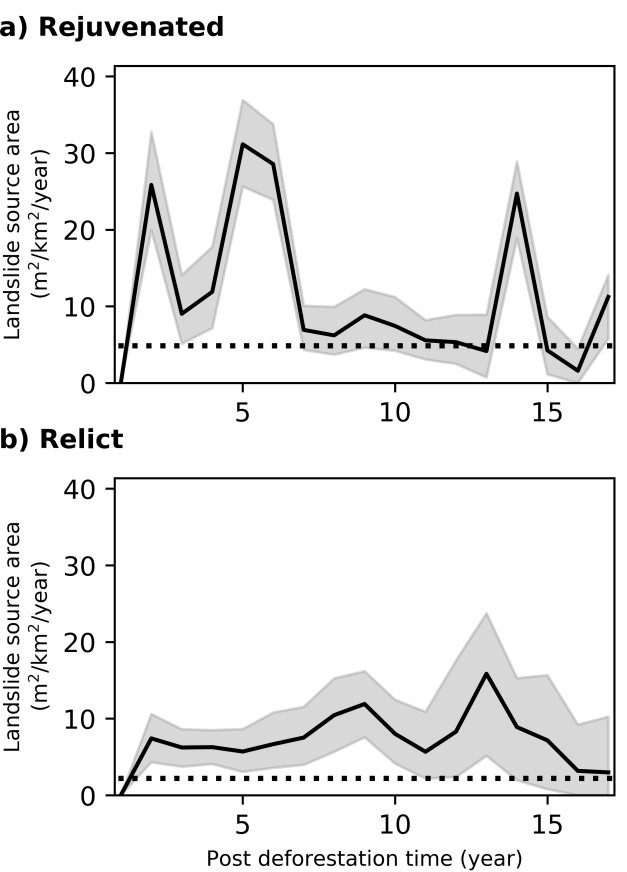

**Figure A1. Deforestation-induced landslide wave in different geomorphic contexts.** Overall landslide source area ($LS_S$, m$^2$ km$^{-2}$ year$^{-1}$) in function of time elapsed since deforestation. The grey area represents the 90 % confidence interval, derived from 100 iterations of $LS_S$ calculations (Section 3.2.4). The dotted line shows the overall erosion rates in the different contexts. **a)** Landslide response to deforestation in rejuvenated landscapes, characterized with 193 landslide instances. The irregular curve suggests that, in order to better characterize the deforestation-induced landslide wave, we need more landslide and deforestation observations and over a larger area. **b)** Landslide response to deforestation in relict landscapes, characterized with 181 landslide instances. The dotted line represents the background erosion rate in relict landscapes.