# Peer review of "Interactions between deforestation, landscape rejuvenation, and shallow landslides in the North Tanganyika - Kivu Rift region, Africa"

_Earth Surface Dynamics, 2020_

## Referee Comment (RC1) · Anonymous Referee #1 · 13 Dec 2020

General comments: This study examined landslide activities in response to deforestation in tectonically-rejuvenated and relict landscapes in the North Tanganyika-Kivu rift region, Africa. The authors mapped landslides from Google Earth imagery using a new method to correct for biases in imagery inventories. They found more abundant but smaller landslides after deforestation in rejuvenated landscapes compared to relict landscapes, which were possibly caused by differences in seismicity and regolith stock. This work tackles an interesting and important topic of how land-use changes affect landslide activities in different geomorphic settings, and has potential to make a contribution to Earth Surface Dynamics. However, the current manuscript could be strengthened with improved data presentation and analyses, and clarification of several key

technical details (detailed below).

Specific comments: Major comments: 1. Landslide definition and data presentation This manuscript termed the mapped landslides as 'shallow landslides' – I don't get what the authors meant /why the authors emphasized 'shallow'? Do the authors refer to shallow soil landslides that are distinct from bedrock landslides, or have no intention to separate soil versus bedrock landslides? Do the authors exclude deep-seated bedrock landslides?

If the mapped landslides are all soil landslides, I can see that landslide size is limited by regolith stock – a recent publication, Prancevic et al. (2020), had a nice dataset illustrating this, which could be a useful reference. If the authors do want to highlight those landslides as 'soil' landslides, some discussions are needed then regarding the possible existence of bedrock landslides in the dataset.

Meanwhile, it would be helpful to display and discuss the areal size distributions (e.g. Malamud et al., 2004) of the landslide inventories in rejuvenated and relict landscapes, which would be a more effective presentation than just mean landslide source area.

2. Ambiguity in the method for correcting image biases The description of the method developed to correct for biases in satellite imagery was confusing, and I could not judge whether this method was correct or not. Specifically, in L165-L170, why is Eq. 6 equivalent to Eq. 7? For example, in Eq. 6, assuming A = 1, N = 3, and n1 = r1 = 3, n2 = r2 = 4, n3 = r3 = 5, I would calculate a LSF of 3. In Eq. 7, if A = 1 and ri > 1, I didn't get how Eq. 7 can give the same result of 3.

I'd suggest add a general paragraph discussing the principles of this correction at the beginning of section 2.2.2, and give some specific examples when doing the derivation. Based on the current description and information, I could not validate this method.

3. Point of ksn analysis I didn't get why the authors introduced a new function relationship between slope and ksn and conducted the analysis in Figure 7 – this seems to be
irrelevant to the characteristics of landslides in rejuvenated vs. relict landscapes, which are the key points of this study. The ksn analysis seems to be isolated from the remaining discussion of landslide activities as well. I'd suggest either removing this part (or moving to supplement) or linking landslide activities to ksn to enrich discussions.

Besides, the classification in Figure 7 was also not convincing – for example, panels a) and f) seem to have similar trends, and do not indicate clearly the existence of a threshold slope (TA).

4. Add analysis of seismicity and rainfall data as controls The authors speculated on the roles of rainfall and seismicity in setting landslide abundance and sizes (section 4.1), but did not conduct any thorough analysis. The authors already presented rainfall data in the region (Figure 2c) which could be analyzed in more details to examine the role of rainfall in landslide occurrence. It'd help as well if more seismicity data could be compiled and analyzed to show the differences in rejuvenated versus relict regions. So I'd suggest more quantitative analysis examining the relationships between rainfall (annual rainfall and rainfall variability or extreme events), seismicity (patterns, numbers of small-magnitude events, etc), and landslide occurrence in the rejuvenated and relict landscapes.

Minor: L1: briefly explain why deforestation increases landslide activity

L6: 'a longer timescale' – over what timescales? Thousands of years?

L7-8: would be useful to define 'rejuvenated' here

L11-L13: too long, consider rewrite as two shorter sentences?

L20: not consistent with discussions and results? Mentioned the role of regolith stock in the discussions and results but didn't mention them here.

L25: clarify what is 'shallow'?

Figure 1: possible to add the spatial extent of landslide mapping? Or the whole area –

would be helpful to indicate.

Figure 2: might help to add a 2-D cross-section plot of the Rift zone and illustrate key parts such as 'shoulders'

L73: is the percentage of tree coverage reported as for each one arc-second grid?

L98: you didn't show the relationships between rainfall metrics and landslide activities in the discussions. . ., also define what is 'sufficiently large'?

L109: modal slope angle?

L125: the introduction of this new function seems somewhat arbitrary, and I really didn't see how this new function adds to the paper. . .

L135-140: do you mean you excluded some really large landslides here? If so, why you want to exclude the large ones? what's the criteria to exclude large ones?

Could expand and add more data to show the results of relative depth? Is there a correlation between landslide area and depth so you can estimate landslide volume?

L140: 'a point of initiation' – is this the centroid point of the source area?

L148: Figure 8a came too late – suggest to move it earlier to Figure 2.

L150: how did you tell landslides from mining and quarrying?

L154: change the title to 'correct for biases in satellite imagery'?

L155: as mentioned earlier, suggest to add an introductory paragraph illustrating the principals of the correction method

L159: the definition of r is not clear. . .maybe give an example to illustrate

L165-170: didn't get why Eq. 6 is equivalent to Eq. 7.

L174: poor definition and explanation of d, again, might help to give an example

L198: in order 'to' link. . .

L216: define what is 'tolerance value of 100 m' here

Figure 7: make it clear in the caption what are the points? Values for each individual catchment?

L233: briefly explain how the classification in Hungr et al. (2014) works?

L253-254: refer to Prancevic et al. (2020)

L266: suggest to make sub-sections in section 4.1 to discuss in each the role of seismicity, rainfall, and regolith stock. . .

Figure 10: in panels g and h, it seems like landslides are centered around similar angles (∼35 degrees) for both rejuvenated and relict landscapes? Why? Similar threshold angles in both landscapes?

L279: what do you mean by 'landslide-triggering earthquakes'? did you mean no earthquakes greater than a specific magnitude? Can make it clear here.

L317: expand and explain what do you mean 'the smaller size is likely due to a smaller minimum critical landslide area linked to the absence of tree cover'?

L322: where does the landslide depth of 2.5 m come from?

L328: how does this 'conservative value of 0.2 mm year-1' relate to Montgomery and Brandon, 2002? Explain.

L341-343: sentence too long and reads confusing. Rewrite to shorter sentences?

References: Prancevic, J. P., M. P. Lamb, B. W. McArdell, C. Rickli, and J. W. Kirchner (2020), Decreasing landslide erosion on steeper slopes in soil-mantled landscapes, Geophysical Research Letters, 47(10), e2020GL087505.

Malamud, B., D. Turcotte, F. Guzzetti, and P. Reichenbach (2004), Landslide inventories and their statistical properties, Earth Surf. Processes Landforms, 29, 687-711.

**ESurfD**

---

## Author Comment (AC1) · 15 Dec 2020

Dear sir/madam,

Thank you for providing this constructive feedback on the research we present in this paper. While a more in-depth response and the updated manuscript will be provided upon receiving the comments of all reviewers, we can already address your main concerns briefly, and point-by-point;

1) Within this work, we opted to look only at shallow landslides, defined here as landslides of which the depth does not exceed more than a couple of meters. We exclude

deep-seated landslides because they are less dependent on direct triggers such as rainfall and processes such as deforestation that are at play at the surface of the earth. We do not expect deforestation to have a (significant) impact on large deep-seated landslides with a rupture plane located deep in the bedrock or regolith of the earth surface. So in conclusion, we only include shallow 'soil' landslides; we exclude deep-seated soil landslides and deep-seated bedrock landslides (shallow bedrock landslides were not observed in the study area). We agree that this can be better highlighted in the text and that the inventory can be further illustrated with the corresponding size-distribution curves.

2) We agree that we used a bit of a shortcut in this part, though the mathematics check out, also for the example you provide; in this example, you describe a database consisting of 12 landslides, of which 3 occur in an area with an imagery range of 3 years, 4 in an area with an imagery range of 4 years, and 5 in an area of imagery range 5 years. To apply Eq. (6), we divide this area into 3 groups (index j); the first group has nj=rj=3, the second nj=rj=4, the third nj=rj=5, hence Eq. (6) gives

LS_F= 3/3+4/4+5/5=3 (assuming A=1).

Now, if we apply Eq. (7) we look at the individual landslides (i) and not the subareas (j). Applying the equation gives

LS_F=1/3+1/3+1/3+1/4+1/4+1/4+1/4+1/5+1/5+1/5+1/5+1/5=3,

the same result, but calculated without grouping the landslides first according to their imagery range (which would yield a very large number of groups in the real world where the imagery range of landslides has continuous values between 0 and 20 years). Perhaps the confusion arises from our switch from summation over subareas j (in Eq. (6)) to summation over individual landslides i (Eq. (7)). We will better describe this step in line 168.

3) In Figure 10 we look at the impact of slope on landslide activity (accounting for

deforestation and rejuvenation). We think that large differences in threshold slope be-tween different areas could have a large impact on the trends observed in this figure; for example, a slope of 30° is expected to display more landslide activity in lithology with a lower threshold slope. Such low-threshold-slope-areas are only present in the rejuvenated landscape. Hence, their presence would have clouded our analysis in Fig. 10: the presence of low-threshold-slope lithology would be an extra factor that could explain the higher landslide rates for slopes in relict landscapes compared to equally steep slopes in rejuvenated landscapes. We will highlight this argument better, and consider moving some of the results towards the supplementary material, considering the comments of the other reviewers.

The classification of the lithology based on threshold slopes was done automatically by investigating whether or not there was a linear relationship between ksn and slope (if R2>0, a linear relationship existed). So while in figure 7a the software detected a positive linear relationship, this was not the case for figure 7f. We will specify this approach in the manuscript. (also note that both lithology a) and f) are excluded from further analysis, since we only focus on the 'strong' lithology).

4) A more thorough analysis of rainfall and seismicity would indeed strengthen this pa-per. Unfortunately, such an analysis is hampered by the data-scarcity in the region. Specifically, to investigate in-depth the impact of rain on landslides, we would need accurate rainfall data (high spatial and temporal resolution) and accurate timing of the landslides (as of now, we can only locate the timing of a landslide somewhere between the age of two Google Earth images). A detailed analysis of rainfall-landslide relation-ships in the region have been conducted by Monsieurs et al. (2020), yet in our case, it is impossible to obtain the necessary rainfall and landslide timing data to complete such an analysis. The same argument can be made for seismicity.

Another possibility is to investigate rainfall-landslide trends is to compare rainfall met-rics (e.g. threshold exceedance/year) with landslide occurrence. Such a study would have merit but has already been conducted for the study area by Depicker et al. (2020).

In that paper, a positive relationship between rainfall and landslides has been established. Similar work has been done for the relation between average seismicity (PGA) and landslides. We will better highlight this previous work and its results in our discussion.

Referenced works: Depicker et al. (2020) The added value of a regional landslide susceptibility analysis: The western branch of the East African Rift. Geomorphology 353 (2020) 106886. https://doi.org/10.1016/j.geomorph.2019.106886

Monsieurs et al. (2019) Towards a Transferable Antecedent Rainfall – Susceptibility Threshold Approach for Landsliding. Water, 11, 2202. https://doi:10.3390/w11112202
* * *

---

## Referee Comment (RC2) · Robert Hilton (Referee) · 22 Feb 2021

Review of Depicker et al., "Interactions between deforestation, landscape rejuvenation, and shallow landslides in the North Tanganyika - Kivu Rift region, Africa" for ESurf.

Please let me apologise for the delay in obtaining the second review of your manuscript. The peer review process is under considerable strain at the moment, we're having to allow reviewers more time. In this case, I have stepped in to act as reviewer (and another AE will take over).

This manuscript seeks to explore how deforestation, and subsequent land use change,

has impacted rates and patterns of landsliding in the North Tanganyika-Kivu Rift region. Landslides are mapped from Google-based satellite imagery across a wide study area (inventory updated from a previous publication) and their spatial and temporal patterns analysed. The paper does a good job of trying to deal with variable bedrock geology (through slope-based correction), and variable image coverage. The results are important – deforestation appears to greatly increase landslide total area and frequency (not a new observation, but novel for this area), but there is an interesting observation about landslide size, which differs in landscapes based on their recent geomorphic disturbance (influence of the rift-associated topographic uplift) (this is new and intriguing). The paper is well set up, and the theme and results are certainly of interest to this field, and the readership at Earth Surface Dynamics.

However, I found that in the abstract, conclusions and discussions, the main findings were not always easy to follow, and that some of the explanations for the patterns could be more convincing.

1) Drawing out the main findings: I found that the main results from the work were quite hard to follow. I wonder if there is a better way to explain what was found, and work through them in the discussion.

In some ways, these key messages can be seen in Figure 9. This firstly shows that deforested landscapes have much larger numbers and rates of landsliding in this setting. I'm aware that other studies have made this point, but I found the discussion of this rather limited.

Second, Figure 9 shows that for all classes, the rejuvenated landscapes have a higher number of landslides. Overall, this translates to an overall greater landsliding area per year. But, in the deforested landscapes the landslides tended to be larger in the relict landscapes, and so covered more total area. This way of explaining the patterns seems a little clearer than the abstract/discussion/conclusion has it.

2) Role of mean annual precipitation: The manuscript refers to a difference in precipitation between the relict and rejuvenated landscapes, but it is rather minor, because in both areas it is close to ∼2 m/yr. I think if a 300 mm/yr difference was seen where the MAP was ∼1 m/yr, this might make more of a difference, but its impact seemed overstated.

The paper also mentions a precipitation intensity proxy. But the output of that analysis was less well explained. It was also unclear how well these climate patterns were constrained for the region, and their spatial variailty across the large mapped area.

3) Landslide maps: The figures are well made, and good on the whole, but the study could benefit from having examples of the landslide polygon maps and spatial patterns of landslide metrics shown in the paper.

Otherwise, my final main comment is similar to point 1 raised by Referee #1 (note I completed my review prior to reading their comments). Namely, the "shallow" part of the landslide description. I don't think this is necessary to add throughout. While the discussion of landslide depth needs to be more complete and considered.

Other comments: 10: at this stage the "relict" and "rejuvenated" terms need more context (perhaps link them to the discussion above).

10 – 13: I have read these two sentence a few times – I find it hard to follow. It starts with "40% higher" erosion rates, and ends with "lower landslide erosion rate".

17: would it make sense to start with these observations (about how deforestation impacts), before then talking about the link to landscape metrics and longer-term patterns (previous text)?

47: Have a new section here, on study area?

76: I suppose there may be reasons why land is "non forest" – i.e. grassland (e.g. precip, temp, seasonal climate) – apart from anthropogenic factors? But perhaps not true in this case, but that could be clarified.

80: a brief summary of some of this would be welcome in the abstract (see comment above).

103: some more information on this climate model output would be welcome. Particularly the link to measurements on the ground, and calibration/testing of model outputs.

108: again, somehow mentioning the attempts to control for lithology could be useful in the abstract (and perhaps mentioned as part of the challenges in this type of assessment in the introduction?)

137: more info is going to be needed on the depth aspect here.

139: what is the normalising area here (km2), and how does it become a per year? – this would benefit from an extra sentence to explain – basically that this is the processing of the landslide maps once each has an initiation year.

148: it would be useful to explain the satellite data source if possible.

170: what is "imagery density"? do you mean more frequent imagery? And/or resolution of imagery? What is the basis to assume that identifying landslides increases linearly with imagery density?

191: I found this section hard to follow. It needs to refer to the necessary equations, and make clear what the issue here is.

208: I see what you're getting at, but this sentence is a bit awkward. Basically you don't know exactly when a landslide occurred between satellite image dates – so just say that.

213: It would be useful to explain for how many landslides this is an issue. Essentially, if the landslide is mapped the same year (or time window) as deforestation. Also, is there not another way to assess this? Could you not examine the values of forest cover pixels proximal to the landslide polygon? If the landslide caused the deforestation, the wider area should still be deforested?

Figure 5 – this is helpful, but it could better draw out the key part of the story (i.e. the middle of part b), perhaps with different annotation?

220: It might be significantly different in a statistical sense, but the value is pretty similar in terms of MAP. . .

221: give the values here.

Figure 6 – the separation of the rejuvenated/relict landscapes is not very clear in this figure. This could be just a display thing (i.e. useful to have an example zoomed in), and/or could be a method thing – how are the knickpoints being used – the non-stationary ones need to have a different symbol.

Figure 8 c – I'm not sure what is happening here – is this across the whole dataset, or a selection of an area, with incremental increases in the image density? The inflection point marked, its not clear to me why this should happen. Is it not because the places with highest image density (fig. 8b) are the cities, with fewer landslides? In other words, is part c the best way to show the role of the image availability on the results?

267: what does the "equal steepness" mean.

269: its not much drier! Both have almost or greater than 2 m of rain per year!

270: in fact, this is about landslide area, not number, so I'm not sure about the explanation here. Why would slightly less rainfall (1.9 m/yr vs 2.2 m/yr) lead to smaller landslides? Is it anything to do with how landslide size is linked to lithology? And/or the geomorphology – are the rejuvenated landscapes typically smaller catchments, and so landslides are generally smaller (constrained by hillslope length)?

Figure 11 – how many landslides? What are the fainter lines and grey zone? What is the landscape average (perhaps as a line?)

290: yes – this makes sense (link to 270 above)

302: That is not what that figure shows, in my opinion – the rate of landsliding seems

to be pretty constant across the window.

306: again, "fading out" is not what I can see in Figure 11.

322: where does this depth come from?! It needs more discussion.

338: I would remove the word "shallow" from here, as you don't actually know how deep they are. Figure A1- this would be better than Figure 11 (and add the number of landslides and explanation of the fine lines and grey shading).

Bob Hilton Durham, UK

---

## Author Response (AR1)

**Response to Reviewing committee comments: Interactions between deforestation, landscape rejuvenation, and shallow landslides in the North Tanganyika - Kivu Rift region, Africa**

**Arthur Depicker**

March 15, 2021

We would like to thank the referees for their detailed and constructive comments. We believe that their feedback identified some weaknesses in our methodology and discussion. Through completing the suggested edits, the revised manuscript benefits substantially from an improvement in the results, overall presentation, and clarity.

To elaborate our answers to the reviewers' comments, the following color scheme is used: comments of the referees are shown in **blue**, answers are annotated in black and quotes from the revised text in **green**. The lines in the final manuscript are indicated in **purple**, while the lines in the manuscript with tracked changes are in **orange**.

**Contents**

| 1        | Referee #1         | 1  |
|----------|--------------------|----|
|          | 1.1 Major comments | 2  |
|          | 1.2 Minor comments | 8  |
| 2 | Referee #2         | 4  |
|          |                    |    |
|          | 2.1 Major comments | 14 |

**1 Referee #1**

General comments: This study examined landslide activities in response to deforestation in tectonically-rejuvenated and relict landscapes in the North Tanganyika-Kivu rift region, Africa. The authors mapped landslides from Google Earth imagery using a new method to correct for biases in imagery inventories. They found more abundant but smaller landslides after deforestation in rejuvenated landscapes compared to relict landscapes, which were possibly caused by differences in seismicity and regolith stock. This work tackles an interesting and important topic of how land-use changes affect landslide activities in different geomorphic settings, and has potential to make a contribution to Earth Surface Dynamics. However, the current manuscript could be strengthened with improved data presentation and analyses, and clarification of several key technical details

**1.1 Major comments**

**1.1a** Landslide definition and data presentation This manuscript termed the mapped landslides as 'shallow landslides' – I don't get what the authors meant /why the authors emphasized 'shallow'? Do the authors refer to shallow soil landslides that are distinct from bedrock landslides, or have no intention to separate soil versus bedrock landslides? Do the authors exclude deep-seated bedrock landslides?

Within this work, we opted to look only at shallow landslides, defined here as landslides of which the depth does not exceed more than a couple of meters. We exclude deep-seated landslides because they are less dependent on direct triggers such as rainfall and processes such as deforestation that are at play at the surface of the Earth. We do not expect deforestation to have a (significant) impact on large deep-seated landslides with a rupture plane located deep in the bedrock or regolith of the earth surface. So in conclusion, we chose to only include shal-low 'soil' landslides; we exclude deep-seated soil landslides and deep-seated bedrock landslides (shallow bedrock landslides were not observed in the study area). We were more explicit in the revised text:

**L158-161/L169-172** Moreover, since deforestation mainly affects the stability of the first few meters of the regolith (Sidle and Bogaard, 2016), we only consider shallow landsliding in this study. Deep-seated and bedrock landslides are excluded from the inventory. We apply a maximum depth of a couple of meters for landslides to be inventoried...

**1.1b** If the mapped landslides are all soil landslides, I can see that landslide size is limited by regolith stock – a recent publication, Prancevic et al. (2020), had a nice dataset illustrating this, which could be a useful reference. If the authors do want to highlight those landslides as 'soil' landslides, some discussions are needed then regarding the possible existence of bedrock landslides in the dataset.

See **comment 1.1a**, we were more explicit about not inventorying deep-seated or bedrock landslides. We added the reference to Prancevic et al. (2020) in the discussion on regolith availability and landslide size.

L322-324/L377-378: ... smaller landslides in comparison to relict landscapes, as the size of the shallow landslides is constraint by the regolith availability (Prancevic et al., 2020).

**1c** Meanwhile, it would be helpful to display and discuss the areal size distributions (e.g. Malamud et al., 2004) of the landslide inventories in rejuvenated and relict landscapes, which would be a more effective presentation than just mean landslide source area.

We illustrate and refer to the size-distribution curves of the landslide inventory, separately for forest and non-forests, and rejuvenated and relict landscapes (there was not enough data to calibrate the curves for deforestation-linked landslides).

Methods:

**L174-176/L196-198**: Furthermore, we illustrate the potential differences between the landslide areas of rejuvenated and relict landscapes by comparing the frequency density of the landslide areas. The frequency density curves are fitted to the inverse  $\Gamma$  distribution (Malamud et al., 2004).

Results:

---

## Author Response (AR2)

April 21, 2021

**Response to the comments of the editor**

Dear editor,

We would like to thank you for providing some additional feedback to polish our manuscript.

Due to the limited amount of comments, among which many are related to spelling or the choice of words, we have addressed your final concerns as comments in the updated track changes file.

Kind regards,

Arthur Depicker & co-authors